# Higher order gaps in the renormalized band structure of doubly aligned hBN/bilayer graphene moiré superlattice

Mohit Kumar Jat[1,5], Priya Tiwari[2,5], Robin Bajaj[1,5], Ishita Shitut[1], Shinjan Mandal [1], Kenji Watanabe [3], Takashi Taniguchi [4], H. R. Krishnamurthy[1], Manish Jain [1] ✉ & Aveek Bid [1] ✉

This paper presents our findings on the recursive band gap engineering of chiral fermions in bilayer graphene doubly aligned with hBN. Using two interfering moiré potentials, we generate a supermoiré pattern that renormalizes the electronic bands of the pristine bilayer graphene, resulting in higher order fractal gaps even at very low energies. These Bragg gaps can be mapped using a unique linear combination of periodic areas within the system. To validate our findings, we use electronic transport measurements to identify the position of these gaps as a function of the carrier density. We establish their agreement with the predicted carrier densities and corresponding quantum numbers obtained using the continuum model. Our study provides strong evidence of the quantization of the momentum-space area of quasi-Brillouin zones in a minimally incommensurate lattice. It fills important gaps in the understanding of band structure engineering of Dirac fermions with a doubly periodic superlattice spinor potential.

Heterostructures of graphene encapsulated between two thin, rotationally misaligned hBN flakes form a stimulating platform for probing topological phases of matter[1-6]. The difference in the lattice constants of hBN and graphene and the angular misalignment between the layers generate two distinct long-wavelength moiré superlattices at the top and bottom interfaces of graphene with hBN[7-11]. The interference between these patterns forms a supermoiré structure with multiple complex real-space periodicities, often with a spatial range larger than that of hBN/graphene moiré at each interface[12-20]. The supermoiré potential (caused by atomic scale modulation of the carbon-carbon hopping amplitudes by the spinor graphene-hBN interaction potential) effectively folds the graphene band over a smaller Brillouin zone while retaining the symmetries of the honeycomb lattice[21]. To first-order, this results in additional, finite-energy split moiré gaps (SMG) in the graphene dispersion[2,7,13,16,22-26]. It was recently realized that the superlattice-induced Bragg reflection at the mini Brillouin zone

boundaries has additional subtler effects on the electronic dispersion of graphene to arbitrary low energies manifested in the formation of a family of Bragg gaps, van Hove singularities, and even possibly flat bands[13,15,27]. Studying these high-order mini-bands and van Hove singularities in graphene/hBN moiré superlattice is essential for a detailed understanding of the emergent quantum properties of quasicrystals[15,28,29] and Dirac fermions in a periodic non-scalar potential[2,16].

Recent momentum-space low-energy continuum model calculations (valid in the low-energy regime of interest[13,30,31]) predict that the positions of these Bragg gaps form a fractal pattern reminiscent of the Hofstadter butterfly[14]. Consequently, the number density of charge carriers at which Bragg scattering (with supermoiré harmonics) occurs can be described by a unique set of Bragg indices (quantum numbers)[14]. These indices, which are integers, relate directly to the filled bands below the gaps and are associated with the quasi-Brillouin

[1]Department of Physics, Indian Institute of Science, Bangalore 560012, India. [2]Braun Center for Submicron Research, Department of Condensed Matter Physics, Weizmann Institute of Science, Rehovot, Israel. [3]Research Center for Electronic and Optical Materials, National Institute for Materials Science, 1-1 Namiki, Tsukuba 305-0044, Japan. [4]Research Center for Materials Nanoarchitectonics, National Institute for Materials Science, 1-1 Namiki, Tsukuba 305-0044, Japan. [5]These authors contributed equally: Mohit Kumar Jat, Priya Tiwari, Robin Bajaj. ✉e-mail: mjain@iisc.ac.in; aveek@iisc.ac.in

Zones (qBZ) formed by the multiple reciprocal lattice vectors of the supermoiré lattice. These indices are topological invariants of the system intimately related to the second Chern numbers[14,32]. Additionally, these minimally incommensurate moiré lattices form an ideal platform to probe the topological properties of quasicrystals. Despite concrete theoretical predictions, this aspect of moiré superlattice remains experimentally unexplored.

Here, we experimentally probe these characteristics of a quasi-periodic lattice using high-mobility heterostructures of bilayer graphene (BLG) doubly aligned with hBN as a model system. From combined measurements of quantum oscillations, longitudinal resistance $R_{xx}$ and transverse resistance $R_{xy}$ of Dirac fermions in this supermoiré potential, we observe and identify a multitude of higher order Bragg gaps and van Hove singularities of the supermoiré structure; these had escaped detection in previous studies[13–18,33]. We map these gaps uniquely to the recently predicted topological Bragg indices of the underlying supermoiré lattice[28]. Furthermore, our continuum modeling of the system shows these zone quantum numbers to have an elegant physical interpretation based on the quantized areas of the qBZ at these Bragg gaps. This model explains the Bragg gaps corresponding to the linear combinations of moiré reciprocal lattice vectors, $p\mathbf{G}_1^b + q\mathbf{G}_2^b + r\mathbf{G}_1^t + s\mathbf{G}_2^t$. Additionally, our analysis explains several unexplained experimental features in graphene/hBN supermoiré systems reported in recent publications[17] (Supplementary Note 8), which were previously studied based on symmetry-based approach[26].

We demonstrate that the BLG supermoiré is different from its single-layer counterpart in several critical aspects – for example, in the symmetry of the moiré Brillouin zone, which has direct consequences for the anomalous Hall effect[34] and electron-electron scattering[35,36], in

terms of the positions and magnitudes of the higher order Bragg gaps (Supplementary Note 11). Additionally, the ability to electrically control the layer and valley degrees of freedom in BLG promises exotic phases that are absent in its single-layer counterpart, e.g. electric field switchable Chern insulators[37].

## Results

### Device characteristics

Heterostructures of BLG doubly aligned with hBN with twist angles less than $0.5°$ were fabricated using the dry transfer technique[38,39] (see Supplementary Note 1). The device is in a dual-gated field-effect transistor architecture, allowing independent control on the charge carrier density $n$ and displacement field $D$ via $n = [(C_{tg}V_{tg} + C_{bg}V_{bg})/e + n_0]$ and $D = [(C_{bg}V_{bg} - C_{tg}V_{tg})/2 + D_0]$ across the device. Here $C_{bg}$ ($C_{tg}$) is the back-gate (top-gate) capacitance, and $V_{bg}$ ($V_{tg}$) is the back-gate (top-gate) voltage. The values of $C_{tg}$ and $C_{bg}$ are determined from quantum Hall measurements. $n_0$ and $D_0$ are the residual charge carrier density and displacement field due to channel impurities, respectively. A plot of the longitudinal resistance $R_{xx}$ measured at $D = 0$ and zero magnetic field is shown in Fig. 1c. The appearance of split moiré resistance peaks at $n_b = \pm 2.36 \times 10^{16}\text{m}^{-2}$ and $n_t = \pm 2.80 \times 10^{16}\text{m}^{-2}$ indicates the alignment of the BLG with both the bottom and top hBN layers. Their presence is also apparent in the 2D map of $R_{xx}$ in the $V_{bg} - V_{tg}$ plane (Fig. 1d).

Figure 1e shows the 2D map of $G_{xx}(n, B)$ at $D = 0$ V/nm in the $n - B$ plane – one finds Landau fans emerging from the charge neutrality point (CNP) and from the secondary Dirac points $n_b$ and $n_t$ with Landau filling $\nu = \pm 4m$ ($m \in$ integer) (Supplementary Note 9). The faint horizontal streaks in the plot are the Brown-Zak oscillations originating from the recurring Bloch states in the superlattice[40,41]. These features get accentuated at high temperatures, where thermal smearing

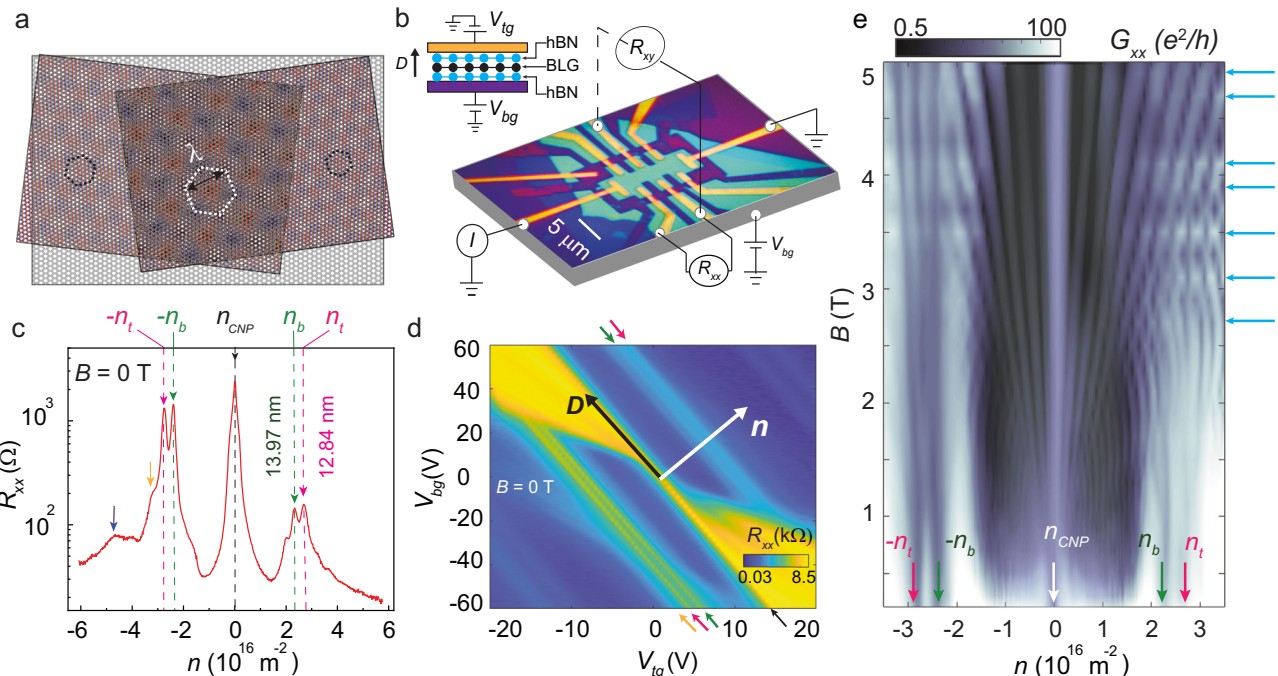

**Fig. 1 | Characteristics of the double moiré device. a** Schematic of doubly aligned BLG with top and bottom hBN. The black and the white hexagons mark the primary moiré and supermoiré plaquettes, respectively. **b** An optical image of the device (before adding the top gate) labeled with the measurement configuration (scale bar: 5 μm). Top inset: Schematic of the layer-stacking, with the direction of increasing displacement field $D$ marked. **c** Plot of the longitudinal resistance $R_{xx}(B = 0)$ as a function of $n$. The black dashed line marks the charge neutrality point. Magenta and dark green lines indicate the secondary Dirac points emerging from top and bottom moiré respectively, with carrier density (moiré wavelength)

$n_t = \pm 2.80 \times 10^{16}\text{m}^{-2}$ ($\lambda_t = 12.84$ nm) and $n_b = \pm 2.36 \times 10^{16}\text{m}^{-2}$ ($\lambda_b = 13.97$ nm), respectively. Yellow and blue arrows mark the higher order Bragg gaps at carrier density $n = -3.3 \times 10^{16}\text{m}^{-2}$ and $n = -4.8 \times 10^{16}\text{m}^{-2}$, respectively. **d** Map of $R_{xx}$ as a function of the back gate voltage, $V_{bg}$ and top gate voltage, $V_{tg}$. The black and white arrows indicate the directions of increasing $D$ and $n$, respectively. The color of the small arrows has the same interpretation as in **c**. **e** Landau-fan diagram $G_{xx}(n, B)$ showing the emergence of Landau levels from the primary Dirac point and the two secondary Dirac points. The cyan horizontal arrows on the right of the plot mark the weak Brown-Zak features. The measurements were done at $T = 2$ K.

diminishes the effect of Landau quantization on the magnetotransport. This is seen clearly from Fig. 2a, which presents the magneto-conductance $\Delta G_{xx}(B)$ plotted in the $n - 1/B$ plane; the data were measured at 100 K. The Fourier transform of a representative data measured at $n = 3.3 \times 10^{16}$ m$^{-2}$ (Fig. 2b) yields multiple frequencies $f = 24.5$T, 29T, and 4T (Fig. 2c). $f$ is related to the real-space area $S$ of the superlattice by $f = \phi_0/S$, where $\phi_0 = h/e$ is the flux quantum[42–45]. The carrier densities (considering two-fold spin and two-fold valley degeneracies) that fill the two first-order moiré Bloch bands are calculated from $f = 24.5$ T and 29 T to be $2.36 \times 10^{16}$ m$^{-2}$ and $2.80 \times 10^{16}$ m$^{-2}$. These number densities match $n_b$ and $n_t$ exactly, identifying these two oscillation frequencies to be associated with the moiré supercell formed at the bottom and top interfaces of BLG, respectively (Supplementary Note 2). The corresponding moiré wavelengths are $\lambda_b = 13.97$ nm and $\lambda_t = 12.84$ nm, respectively. The Brown-Zak frequency $f_s = 4$ T yields $n_s = 0.39 \times 10^{16}$ m$^{-2}$ – this number density corresponds to a real-space wavelength of $\lambda_s = 34.6$ nm which is the size of the super-moiré unit cell in our heterostructure (Supplementary Note 2). We thus identify $f_s$ to be the supermoiré Brown-Zak frequency.

To verify that the split peaks at $n_b$ and $n_t$ are not artifacts due to large angle-inhomogeneity in the device, we repeated the measurements on a control device (labeled D$_{single}$) where only the top-hBN forms a moiré with the BLG (Supplementary Note 2). To achieve this, a single-layer WSe$_2$ was interposed between half of the BLG and the lower hBN. The $n - R_{xx}$ plot of this single-moiré device had a single secondary peak at $n = n_t$ (See Supplementary Fig. 3). This helps ascertain that both the top- and bottom-hBN crystals have the same relative rotation direction with the intervening graphene layer for the double-moiré device, with twist angles $\theta_b = 0.03° \pm 0.03°$ between bottom hBN and graphene and $\theta_t = 0.44° \pm 0.03°$ between top hBN and graphene (Supplementary Note 2). The very small values of the twist angles place our device in the commensurate limit[46].

## Continuum Hamiltonian

Having established the presence of the supermoiré structure, we move on to discuss its effect on the bilayer graphene band structure using the Bistritzer-MacDonald continuum model[47]. The $4 \times 4$ effective Hamiltonian (eliminating the sub lattice basis of hBN using second-order perturbation theory) is written as:

$$H_{eff} = \begin{bmatrix} H_G + V_{hBN}^b & U_{BLG}^\dagger \\ U_{BLG} & H_G + V_{hBN}^t \end{bmatrix} \tag{1}$$

where, in the low-energy limit,

$$V_{hBN}^\ell = U^{\ell\dagger}(-H_{hBN})^{-1}U^\ell = v_0 + v_1 e^{i\xi \mathbf{G}_1^\ell \cdot \mathbf{r}} + v_2 e^{i\xi \mathbf{G}_2^\ell \cdot \mathbf{r}} + v_3 e^{i\xi \mathbf{G}_3^\ell \cdot \mathbf{r}} \tag{2}$$

Here $\ell = t, b$ and $\xi = \pm 1$ is the valley index. $\mathbf{G}_1^\ell$ and $\mathbf{G}_2^\ell$ are the reciprocal lattice vectors of the $\ell$ moiré and $\mathbf{G}_3^\ell = -\mathbf{G}_1^\ell - \mathbf{G}_2^\ell$. $U_{BLG}$ is the inter-layer potential between the layers of the BLG.

Figure 3a shows the theoretically constructed density of states (DOS) versus carrier density plot; the zeros in the DOS correspond to the gaps in the energy spectrum. To gain a physical understanding of the origin of these gaps, we follow the procedure laid out in ref. 14. Recall that a nearly commensurate system with dual periodicity is defined by a set of four distinct reciprocal wave vectors: $\mathbf{G}_{1,2}^t$ being the two primitive reciprocal lattice vectors of the moiré lattice at the top hBN-graphene interface and $\mathbf{G}_{1,2}^b$ those for the second moiré lattice at the bottom graphene-hBN interface. One can form quasi-Brillouin zones bounded by multiple Bragg planes defined by a linear combination of these four primary reciprocal vectors. The $(m_1, m_2, m_3, m_4)^{th}$ – order Bragg-gap appears in the electronic spectrum when the total

charge carrier density equals[14,33]:

$$n(m_1, m_2, m_3, m_4) = 4 \sum_{i=1}^{4} m_i A_i / (2\pi)^2. \tag{3}$$

Here $A_1 = |\mathbf{G}_1^b \times \mathbf{G}_2^b|$, $A_2 = |\mathbf{G}_1^t \times \mathbf{G}_2^t|$, $A_3 = |\mathbf{G}_1^b \times \mathbf{G}_2^t|$, and $A_4 = |\mathbf{G}_1^t \times \mathbf{G}_2^b|$ are the areas of the projections of the parallelograms formed by the four reciprocal lattice vectors $\mathbf{G}_i$. The quantity $\sum_{i=1}^{4} m_i A_i$ is the area (in reciprocal space) of the multifaceted quasi-Brillouin zone, and the factor of four on the right-hand side of Eq. (3) arises from the spin and valley degeneracies. The areas $A_i$ for the experimentally obtained twist angle $\theta_b = 0.03°$ and $\theta_t = 0.44°$ are 0.277, 0.233, 0.181 and 0.290 nm$^{-2}$. The integers $m_i$ are Bragg indices (quantum numbers) of the gap and are topological invariants of the system[14,28]. Note that this formalism is mathematically identical to that utilized by previous workers based on differences between the multiples of the aligned and rotated reciprocal vectors[13,15,17,33] with the added advantage of being intuitively transparent.

Note that in our theoretical calculations, we used the lattice constant of graphene and hBN to be 0.246nm and 0.2504nm, respectively (corresponding to a strain, $\epsilon = 0.018$). We also considered other values of the strain parameter in the commonly used range $0.0165 \leq \epsilon \leq 0.0185$. The theoretical values generated with $\epsilon = 0.018$ match best with our measured experimental results.

## Experimental observation of the Bragg gaps

Using the above formalism, the band gaps corresponding to the densities $n_b$, $n_t$, and $n_s$ are identified to be Bragg gaps with Bragg indices $(0, 1, 0, 0)$, $(1, 0, 0, 0)$, and $(1, 1, \bar{1}, \bar{1})$ respectively (see Supplementary Note 5). We obtain the positions of additional Bragg gaps by comparing calculated DOS (Fig. 3a) with the experimentally determined transverse resistance $R_{xy}(B)$ and the extracted Hall carrier density $n_H = B/(eR_{xy})$ measured in the presence of a small, non-quantizing magnetic field $B = 0.7$T (Fig. 3b, c).

The zeroes (and several prominent non-zero dips) in the calculated DOS are reflected in the experimental data as a discontinuity in the $n_H - n$ plot. Recall that in a multi-carrier type system and for small $B$, a change in sign of $R_{xy}$ (or a corresponding divergence in $n_H$) can either indicate a bandgap or a van Hove singularity[48,49]. The sign of $n_H$ on either side of a band gap reflects the local band curvature (and hence, the carrier type). Thus, for instance, with $E_F > 0$, one can have both positive and negative $n_H$; a positive (negative) value of $n_H$ implies an electron-like (a hole-like) band (we take the electronic charge to be $e$). A band gap can be said to exist at a certain number density if the following three conditions are simultaneously met: (1) the DOS in Fig. 3a goes to zero, (2) $n_H$ in Fig. 3c changes sign, and (3) there is a local maximum in the $d^2 G_{xx}/dn^2$ (minima in $G_{xx}$) data in Fig. 3d. Using this criterion, we identify the principal gaps at $n_b = -2.36 \times 10^{16}$ m$^{-2}$, $n_t = -2.80 \times 10^{16}$ m$^{-2}$, and $n_{CNP} = 0$ m$^{-2}$ as Bragg gaps with quantum numbers $(0, \bar{1}, 0, 0)$, $(\bar{1}, 0, 0, 0)$, $(0, 0, 0, 0)$, and respectively. We also identify several higher order Bragg gaps, for example, at $n = -3.3 \times 10^{16}$ m$^{-2}$ $(2, 2, \bar{1}, \bar{4})$ and $-4.8 \times 10^{16}$ m$^{-2}$ $(\bar{4}, \bar{1}, 0, 3)$. We mark all the Bragg gaps with solid gray lines in Fig. 3a–f.

There are certain number densities for example, at $-0.39 \times 10^{16}$ m$^{-2}$ $(\bar{1}, \bar{1}, 1, 1)$ and $-6.6 \times 10^{16}$ m$^{-2}$ $(\bar{1}, 2, \bar{3}, \bar{1})$, (marked by dotted blue lines in Fig. 3a–d) where the DOS goes to zero and the $G_{xx}$ has a minima, however $n_H$ does not reach zero. We tentatively identify them as narrow Bragg gaps that are masked by thermal/impurity broadening. Note also that there are no gaps at positive energies with the exception of the supermoiré gap at $n_s = 0.39 \times 10^{16}$ m$^{-2}$ $(1, 1, \bar{1}, \bar{1})$. The reason why the supermoiré gap survives the band overlaps (that quenches all other gaps at positive energies) is at present unclear.

Additionally, there are features at which $n_H$ changes sign accompanied by minima in $d^2 G_{xx}/dn^2$ (maxima in $G_{xx}$) and a peak in DOS – we identify these to be due to van Hove singularities. Two of these

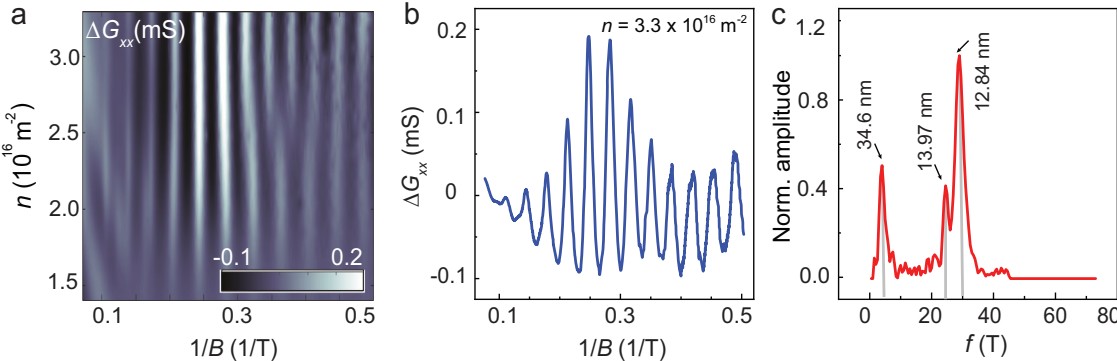

**Fig. 2 | Brown-Zak oscillation in double moiré device. a** Brown-Zak oscillations of magnetoconductance $\Delta G_{xx}$ plotted in the $n-1/B$ plane; the data were measured at $T=100$ K. **b** Plot of $\Delta G_{xx}$ as a function of $1/B$ for carrier density $n=3.3\times10^{16}\mathrm{m}^{-2}$. **c** The Fourier spectrum of the data in **b**, shown as a function of $f$(T); the peaks are marked with the corresponding moiré super-lattice wavelengths.

**Fig. 3 | Experimentally obtained and theoretically calculated Bragg gaps. a** Plot of the calculated density of states (DOS) for $\theta_b=0.026°$ and $\theta_t=0.44°$. **b** Plot of transverse resistance $R_{xy}$ versus $n$ measured at $B=0.7$ T and $T=2$K. **c** Plot of Hall carrier density $n_H$ versus $n$. **d** Map of the normalized $d^2G_{xx}(B=0)/dn^2$ in the $n-D$ plane; the $d^2G_{xx}(B=0)/dn^2$ have been plotted on a logarithmic scale. The indices of the Bragg gaps are marked on the right. In **a–d**, the solid gray lines mark the values of $n$ at which the Bragg gaps open with DOS = 0, $R_{xy}=0$ and $d^2G_{xx}(B=0)/dn^2$ having a maxima; the dotted blue lines mark the values of $n$ at which the Bragg gaps open with DOS = 0 and $d^2G_{xx}(B=0)/dn^2$ having a maxima, but $R_{xy}$ does not reach zero. **e, f** Zoomed-in plots of DOS and $n_H$ versus $n$ in a narrow range on the hole-side. The solid gray lines indicate the Bragg gaps, while the dotted purple lines indicate the locations of the van Hove singularities. **g** Plot of the position of a few representative Bragg gaps versus $n$ over a range of electric fields. The corresponding quantum numbers of the gaps are marked on the right.

(at $n=-2.05\times10^{16}\mathrm{m}^{-2}$ and $n=-2.6\times10^{16}\mathrm{m}^{-2}$) have been marked with purple dotted lines in Fig. 3e, f. Note that, in addition to the ones marked, the calculated DOS plotted in Fig. 3a shows several dips at which the measured longitudinal and Hall resistances are featureless. We find that at these points, either the DOS is finite with no band gap,

or the calculated band gaps $\Delta$ are substantially smaller than 1meV (e.g. at $n=-2.18\times10^{16}\mathrm{m}^{-2}$, $\Delta=0.74$ meV) and hence not resolvable in our electrical transport measurements.

The fact that the data from measurements of three independent physical quantities (quantum oscillations, Hall resistance, and

zero-magnetic-field longitudinal resistance) and from continuum-model-based calculations match emphasizes the validity of our analysis. We note in passing that the positions of the primary gaps in number density are independent of applied small displacement fields (Fig. 3g).

From the activated temperature-dependent resistance data, we extract the band-gap at CNP to be 6meV at zero displacement field. This value is in the same range as our theoretically calculated band gap 4meV and is in agreement with the recent theoretical work in supermoiré system[31] and experimental studies in transport[19]. The energy gaps at the primary moiré gaps on the hole-side are extracted to be $\Delta_b = 1.46$ meV and $\Delta_t = 3.39$ meV respectively.

## Quasi Brillouin Zones

The electronic carrier densities at which we observe the gaps in our doubly-periodic 2D system are related to the areas of the underlying qBZ. In order to identify these zone boundaries, we find the **k**-points at which the gaps open. One can observe the gap opening points by unfolding the supermoiré band structure to the unit cell of the BLG. We modulate the strength of top and bottom moiré potential in the reduced Hamiltonian (Eq. (1)) with strength parameter $\eta$ ranging from 0 to 1 (See Supplementary Note 6). The unfolded band structure (Fig. 4a) can be seen along a given **k**-path using unfolded spectral weights as:

$$A(\mathbf{q}, \epsilon) = \sum_{n\mathbf{k}} \sum_{X} |\langle \mathbf{q}, X | \psi_{n\mathbf{k}} \rangle|^2 \delta(\epsilon - \epsilon_{n\mathbf{k}}) \tag{4}$$

where $X = A^1, B^1, A^2, B^2$ denote the atoms of bilayer graphene, $|\psi_{n\mathbf{k}}\rangle$ and $\epsilon_{n\mathbf{k}}$ denote the eigenstates and eigenvalues, respectively, $\mathbf{q}$ is the crystal momentum in the bilayer graphene unit cell BZ. The $\mathbf{q}$ is related to **k** in the supermoiré BZ with a moiré reciprocal lattice vector $\mathbf{G}_{SM}$ via the relation $\mathbf{q} = \mathbf{k} + \mathbf{G}_{SM}$[21].

Figure 4b–e shows the calculated qBZ for a few Bragg indices using the above procedure. These shapes and the corresponding Bragg indices have simple geometrical interpretations. Consider, for example, the qBZ of the supermoiré cell plotted in Fig. 4c; it is formed by the reciprocal lattice vectors $\mathbf{G}_1^b - \mathbf{G}_1^t$ and $\mathbf{G}_2^b - \mathbf{G}_2^t$. The area of this

qBZ can be expressed as:

$$(\mathbf{G}_1^b - \mathbf{G}_1^t) \times (\mathbf{G}_2^b - \mathbf{G}_2^t) = (\mathbf{G}_1^b \times \mathbf{G}_2^b) + (\mathbf{G}_1^t \times \mathbf{G}_2^t) - (\mathbf{G}_1^b \times \mathbf{G}_2^t) - (\mathbf{G}_1^t \times \mathbf{G}_2^b)$$
$$= |\mathbf{A}_1| + |\mathbf{A}_2| - |\mathbf{A}_3| - |\mathbf{A}_4| \tag{5}$$

This gives the Bragg indices of the qBZ of the supermoiré to be $(1,1,\bar{1},\bar{1})$ (see Eq. (3)) with the number density required to fill the band $n_s = 0.39 \times 10^{16}\,\mathrm{m}^{-2}$. We thus find the area of the supermoiré qBZ arrived at using two very different theoretical routes (continuum model calculations and band geometric considerations) to be in excellent agreement with that extracted from measured Brown-Zak oscillations.

A closer inspection reveals that several of the qBZ are three-fold symmetric; two examples are provided in Fig. 4c, d. The source of this $\mathcal{C}_3$ symmetry can be traced back to the triangular symmetry of the constant energy contours of bilayer graphene energy dispersion (See Supplementary Note 7). Figure 4e shows an example of the fractal or flower-like qBZ for higher order gap Bragg predicted for doubly aligned graphene[14].

## Discussion

We note in passing that throughout the above discussion, we have avoided any mention of the strength of the interlayer coupling. As noted in previous studies, the interlayer coupling strength affects only the magnitude of the Bragg gaps, leaving their positions unaffected[13].

To summarize, we have shown that the low-energy dispersion of bilayer graphene can be significantly altered by the supermoiré potential. Our study provides an elegant physical picture of the Bragg gaps opening in the moiré spectrum (based on area quantization of the qBZ) and helps identify the relevant topological quantum numbers. Our experimental results match very well with the predictions of the subtle effects of nearly commensurate supermoiré structures on graphene bands. Importantly, our calculations establish that the qBZ of the supermoiré lattice in bilayer graphene are $\mathcal{C}_3$ symmetric (in contrast to single-layer supermoiré), making it an ideal system to host intrinsic Berry curvature dipoles. The scope of topology has been limited to strictly periodic systems, but our study represents a crucial step toward expanding it to encompass quasicrystals and their topological properties. To fully comprehend the physics of these intriguing materials and unlock their complete potential, additional experiments

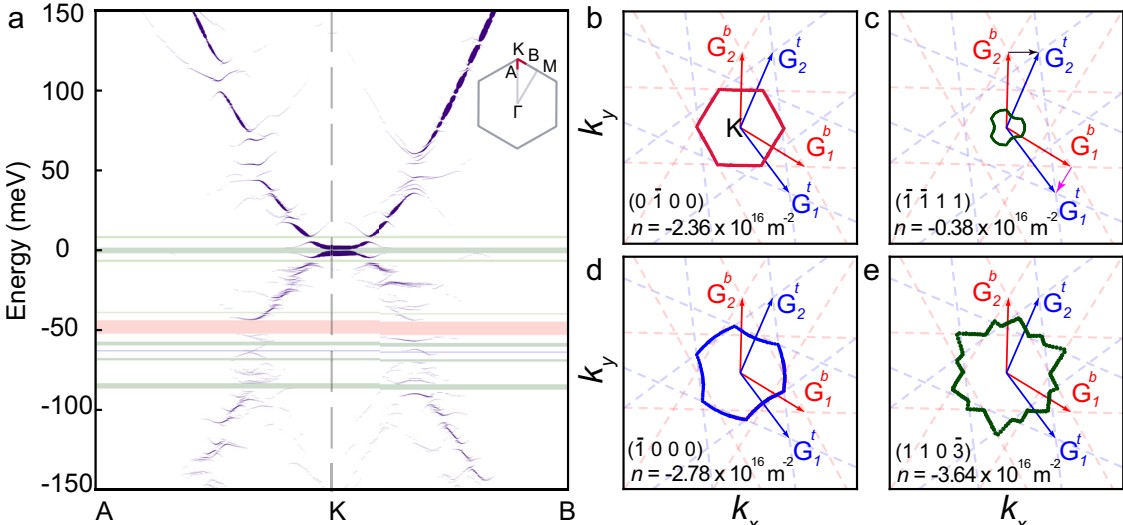

**Fig. 4 | Unfolded band structure and qBZs. a** Unfolded band structure along the path AKB (shown in the subplot). Primary gaps from the top and bottom moiré are shown in red and blue. Gaps arising from both layers, i.e. supermoiré gaps, are shown in green. **b–e** Plots of the calculated qBZ for the Bragg gaps at number densities −2.36 × 10¹⁶ m⁻², −0.38 × 10¹⁶ m⁻², −2.78 × 10¹⁶ m⁻², and −3.64 × 10¹⁶ m⁻², respectively. The Bragg indices are indicated in the insets of each panel. The k-points where the gap opens are shown as dots in red, blue and green. The $k_x$ and $k_y$ range between [−0.8,0.8] nm⁻¹.

and theoretical calculations that incorporate interaction effects are necessary.

## Methods

### Device fabrication

Devices of bilayer graphene (BLG) heterostructures doubly aligned with single crystalline hBN were fabricated using a dry transfer technique (for details, see Supplementary Note 1). Flakes of hBN and bilayer graphene were exfoliated on the Si/SiO$_2$ substrate with a thickness of 280nm. Raman spectroscopy and AFM were used to determine the number of layers and thickness uniformity, respectively. A poly-dimethylsiloxane (PDMS) dome coated with a sacrificial polycarbonate (PC) layer was used to pick up the flakes sequentially. The rotation stage was coupled with the 3D stage manipulator to control the distance and angle between the flakes independently. The heterostructure was aligned under the microscope to form a moiré superstructure with less than 1° misalignment. The final constructed device was vacuum annealed at 280 °C for 10 h. The devices were patterned using standard electron beam lithography, followed by reactive ion etching (using mixture of CHF$_3$ (40sccm) and O$_2$ (4sccm)) and thermal deposition of Cr/Au (5 nm/55 nm) contacts. The dual-gated device architecture allows for independent tuning of charge carrier density and displacement field. The capacitance values of the top gate and back gate were extracted from quantum hall measurements.

### Transport measurements

Electrical transport measurements were performed in a cryogen-free refrigerator (with a base temperature of 2 K and magnetic field up to 14 T). These measurements were performed at low frequency (18.8 Hz) using standard low-frequency measurement techniques at a bias current of 10 nA.

### Uncertainty in twist angle estimation

The twist angle is estimated using the relation

$$n = \frac{8[\epsilon^2 + 2(1+\epsilon)(1-\cos(\theta))]}{\sqrt{3}a^2(1+\epsilon)^2} \tag{6}$$

Here, $n$ is carrier density corresponding to the fully filled superlattice unit cell, a = 0.246 nm is the lattice constant of graphene, $\epsilon$ = 0.018 is the lattice mismatch between the hBN and graphene, and $\theta$ is the relative rotational angle between the two lattices. Uncertainty in the twist angle ($\delta\theta$) is estimated from the uncertainty in the carrier density ($\delta n$) by the following relation:

$$\delta\theta = \frac{\sqrt{3}\,a^2(1+\epsilon)}{16\,\sin(\theta)}\delta n \tag{7}$$

The impurity carrier concentration from the transport measurement is extracted to be $\delta n = 7 \times 10^{14} \mathrm{m}^{-2}$. At the twist angle of $\theta \approx 0.5°$, uncertainty in the twist angle is extracted to be $\delta\theta = 0.03°$.

## Data availability

The authors declare that the data supporting the findings of this study are available within the main text and its Supplementary Information and at https://doi.org/10.6084/m9.figshare.25195817. Other relevant data are available from the corresponding author upon request.

## Code availability

The code that support the findings of this study are available from the corresponding author upon request.

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

## Acknowledgements

The authors acknowledge Simrandeep Kaur for help with device fabrication. A.B. acknowledges funding from U.S. Army DEVCOM Indo-Pacific (Project number: FA5209 22P0166) and Department of Science and Technology, Govt of India (DST/SJF/PSA-01/2016-17). M.J. and H.R.K. acknowledge the National Supercomputing Mission of the Department of Science and Technology, India, and the Science and Engineering Research Board of the Department of Science and Technology, India, for financial support under Grants No. DST/NSM/R&D_HPC Applications/ 2021/23 and No. SB/DF/005/2017, respectively. M.K.J. and R.B. acknowledge the funding from the Prime Minister's research fellowship (PMRF), MHRD.

## Author contributions

M.K.J., P.T., and A.B. conceived the idea of the study, conducted the measurements, and analyzed the results. T.T. and K.W. provided the hBN crystals. M.J., R.B., S.M., I.S., and H.R.K. developed the theoretical model. All the authors contributed to preparing the manuscript.

## Competing interests

The authors declare no competing interests.
