## [Peer Review File · Nature Communications]

REVIEWER COMMENTS

Reviewer #1 (Remarks to the Author):

- The authors present a clean work combining experimental results and theory to quantify the superlattice effects in hBN-encapsulated BG. The physics at play are already well understood from a theoretical point of view where the authors chose the cleanest option available to interpret the results from a geometrical perspective matching the areas of the quasi BZ with the corresponding densities at which the superlattice features are expected to occur, but the novelty in their work is to apply it on bilayer graphene whereas all current works that I am aware of the focus lies on single-layer graphene.

I am not sure this limited novelty warrants publication in Nature Communications unless the authors can provide some deeper insights as to how their bilayer graphene results differ and agree with the existing observations in single-layer graphene. For instance, can you go into more detail about the sudden claim in the conclusion that the qBZs in bilayer graphene are C_3 symmetric? Is this different from single-layer graphene? Are there any other effects that are expected to happen solely in BG but not in SL?

The above paragraph is by no means a critique of the quality of the work; it's just the novelty part that I am not fully sure about.

I have a few additional comments that the authors should address:

- Can you comment on why the electron gap is non-zero, while in BG on hBN (without encapsulation) the gap is expected to be zero due to band overlap [see Nano Lett. 2018, 18, 12, 7732–7741 (2018)]. Do you have a band structure for zero-degree aligned hBN-encapsulated BG to visualize this behavior and see how the encapsulation opens up this gap? You can also do it at finite angles if you find a small enough commensurate cell for the double moire system but the zero-degree alignment should already provide some information.
- Do you know if the hBN layers have their B and N atoms on top of each other respectively for 0-degree alignments or the B atom is on top of the N atom and the N atom is on top of the B atom, corresponding to a 60 degree rotation of the top layer?
- Presumably being in the commensurate limit, how does this affect your estimates of the twist angles based on the moire length?
- Similarly, how does potential strain affect your angle estimates?
- Small comment 1: f is at some point referred to as a frequency, later on as a periodicity. Can you remove this ambiguity?
- Small comment 2: Can you define f_s the first time you introduce it in the text (maybe I missed it)?

Reviewer #2 (Remarks to the Author):

- The authors study the moire superlattice effect in the electronic structure in bilayer graphene doubly aligned with top and bottom hBN layers. By the electronic transport and magneto-transport measurements, they observed a number of Bragg gaps opening in the low-energy spectrum, which are confirmed to quantitatively agree with

the theoretical calculation using a continuum model.

As a major finding of the paper, they identified the topological numbers and associated quasi-Brillouin zones for major gaps

using the theoretical approach proposed in Ref. [13].

This is the first work that experimentally validates

the topological gap labeling in quasi-crystalline systems,

and I find it is potentially worth publication in Nature Communications.

I request the authors to address the following problems, concerning

the correspondence between the experimental results and the theoretical calculation.

--- In Fig.2f, the Hall carrier density n_H diverges at five points.

I understand each of them corresponds to a band gap, and

the sign of n_H indicates the carrier type (electron/holes).

When going down from the Dirac point, for instance,

n_H diverges to the negative infinity at $n \sim -2.2 \times 10^{16}/\text{m}^2$

and then it comes back from positive infinity,

suggesting that we have electron bands and hole bands

above and below the gap, respectively, as naturally expected.

However, the behavior becomes opposite in the next diverging point ($n \sim -2.4 \times 10^{16}$);

i.e., we have holes above and electrons below the gap, and it's alternating for the 3rd, 4th and 5th.

This is an interesting part, but it's hard to imagine from the band calculation,

because we naively expect a band bottom above the gap, and a band top below the gap.

Can the band calculation simulate this behavior?

We also should have the van-Hove singularity somewhere in the middle of the band,

at which the carrier type changes while not diverging.

Also, the four-number gap labeling is missing

for the fourth diverging point ($n = -2.5$),

while the divergence behavior is quite pronounced.

Any possibility to have a big gap between two main gaps?

Based on these considerations, I remain somewhat unconvinced about the agreement between the experimental results and the theoretical calculations.

--- In Fig. 2d, the gap structure does not sensitively depend on the displacement field D . Does this feature agree with the theoretical band calculation?

Generally the band structure of AB-stacked graphene is strongly modified by the perpendicular electric field.

In increasing D , I imagine the gap at zero energy (0,0,0) increases and other minigaps would also be significantly modified.

Some minor issues:

--- The correlation between the text and figures is not very cohesive and

it is sometimes hard to relate.

For instance, the n_b , n_t , etc should be indicated in fig.1 and fig.2.

Also it would be convenient if the four numbers are also indicated in fig.2e and 2f.

--- What do gray solid/dashed/dotted lines represent in Fig. 2?

--- The Brown-Zak frequency f_s seems mentioning $4T$ but not defined in the text.

--- Is θ_b fixed to 0.026 deg in model p1 to p4 in Sec. S4?

Reviewer #3 (Remarks to the Author):

See attachment.

The study of moiré superlattices has been emerging in the past few years, yet there are still many open questions in this field. One of the perplexing problems is the moiré quasiperiodicity, or a system with two moiré wavelengths that interfere. This manuscript studies a bilayer graphene double aligned with top and bottom hBN, and observes supermoiré patterns which can be explained by a simple picture of linear combination of periodic areas. This analysis bridges a gap in this field, which I believe will be of general interest. However, I think the data quality is below standard and some of the data interpretation is not convincing enough. Especially, the raw experimental data is too scarce that cannot fully support the argument. Therefore, I don't think it meets the criteria of Nature Communications. Below are my comments.

1. I don't see the superlattice peak of supermoiré. Do the authors observe it in the resistance measurement? If not, what might be the reason that they don't see it?
2. The authors should present the raw data of Landau fan diagram for readers to get more information of the device.

The Brown-Zak oscillation data is too vague and it's hard to tell more than one oscillation period by observing the 2D map. At least the authors should present a linecut at specific doping before presenting its the Fourier spectrum in Fig.1f.

3. In Fig. 1d, there are a few solid lines and dashed lines for guiding, but these lines are too thick that I don't even see the features beneath them. Also, what are the dashed lines referring to?
4. In Fig.2, the authors compare the Hall measurement with calculated DOS. However, there are too many lines and it's difficult to locate where the authors are referring to as "additional peaks" or "main features". Both the notations in figures and in the manuscript should be clearer. There should be a baseline of "0" drawn in the R_{xy} and n_H figures for easier discussion. Some of the features in DOS, R_{xy} and n_H do match, but there are many features in the DOS. For example, the following line pointed by red doesn't necessarily correspond to a gap feature in DOS. Also, why are there some other prominent dips in DOS not shown as peaks in R_{xx} or divergence in R_{xy} , such as the one pointed by blue?

5. The authors mention that the system is a ferroelectric phase, so does it have hysteresis when scanning either gate backward and forward? Does it having "layer-specific anomalous

screening" behavior as discussed in *Nature* **588**, 71–76 (2020)? Which gate sweeping direction did the authors choose for their dual-gate scan? Although the ferroelectricity and supermoiré pattern can be independent, the authors should present the details of the raw experimental data instead of claiming it showing ferroelectricity without illustrating any details.

6. Is there any specific reason for using bilayer graphene for the study? Do the authors expect anything different if they use a monolayer graphene (e.g. *Science Advances* **5**, eaay8897 (2019) used a monolayer for similar studies)?
7. In the supplementary, the authors reproduce the data of *Science Advances* **5**, eaay8897 (2019), but one of the data points should be $+4.1 \times 10^{12}/(\text{cm}^2)$ instead of $+4.35 \times 10^{12}/(\text{cm}^2)$ in the original paper. I think this is why the line for guidance doesn't even correspond to a peak in R_{xx} in the supplementary (pointed by red).

The Reviewers' comments are shown in bold, and our responses, inserted after each specific point, are in regular font. This is followed by a list of significant changes made in the manuscript.

REPORT OF REVIEWER 1 NCOMMS-23-19425-T

Comment 1: The authors present a clean work combining experimental results and theory to quantify the superlattice effects in hBN-encapsulated BG. The physics at play are already well understood from a theoretical point of view where the authors chose the cleanest option available to interpret the results from a geometrical perspective matching the areas of the quasi BZ with the corresponding densities at which the superlattice features are expected to occur, but the novelty in their work is to apply it on bilayer graphene whereas all current works that I am aware of the focus lies on single-layer graphene.

Response: We thank the Reviewer for their appreciation of the work.

Comment 2: I am not sure this limited novelty warrants publication in Nature Communications unless the authors can provide some deeper insights as to how their bilayer graphene results differ and agree with the existing observations in single-layer graphene. For instance, can you go into more detail about the sudden claim in the conclusion that the qBZs in bilayer graphene are C3 symmetric? Is this different from single-layer graphene? Are there any other effects that are expected to happen solely in BG but not in SL? The above paragraph is by no means a critique of the quality of the work; it's just the novelty part that I am not fully sure about.

Response: We agree with the Reviewer that the motivation behind studying bilayer-based supermoiré was not brought out clearly in our original manuscript. As the Reviewer pointed out correctly, single-layer and bilayer graphene-based supermoiré systems have certain similarities, principal among them are: (1) for a given twist angle, the moiré wavelength and the number densities at which Bragg gaps open are the same for both systems and (2) the concept of higher order fractal gap applies to both systems. However, as we list below, there are significant differences between these two material platforms.

1. As we show in this letter, the shapes of the Brillouin zone are very different for the two systems (Fig. 1 of this document). For the bilayer graphene supermoiré, the Brillouin zone is a trigonally distorted hexagon with C_3 symmetry, which is a consequence of intrinsic trigonal warping in pristine bilayer graphene. On the other hand, for single-layer graphene supermoiré, the Brillouin zone is hexagonal. (Figs. S10 and S11 of the Supplementary Materials).
2. This difference directly affects the umklapp scattering in these systems. The umklapp threshold carrier density is independent of twist angle for single-layer graphene moiré [1], and it becomes a non-monotonic function of twist angle for bilayer graphene moiré system [2], which is a consequence of the trigonally warped Brillouin zone.
3. Unlike single-layer graphene supermoiré, bilayer graphene supermoiré systems are predicted to host electric field switchable Chern insulators [3].
4. Experimental tunneling spectroscopy studies suggest that in an SLG single moiré, a bandgap appears at the secondary Dirac points, while the BLG single moiré is gapless at all number densities [4]. Thus it is instructive to compare the bands of supermoiré in SLG and BLG to check if the same distinction persists. Our studies establish that this is not the case – supermoiré opens a gap at the secondary Dirac point in BLG (in contrast to single moiré in BLG that fails to open a bandgap at the secondary Dirac point).
5. In the doubly aligned bilayer graphene supermoiré system, there are recent reports of observation of a ferroelectric phase (arising presumably from electric field-induced layer polarization) [5–7]. This physics is not possible in the doubly aligned single-layer graphene system. The ability to electrically tune the layer and valley degrees of freedom in BLG supermoiré makes it quite distinct from SLG supermoiré.

FIG. 1. **BZs for SLG and BLG aligned with hBN.** Brillouin Zones (BZ) constructed by unfolding the primary moiré gaps for a twist angle of $\theta_t = 0.44^\circ$. The blue plot show the six-fold symmetric BZ of single-layer graphene/hBN moiré. The green plot shows the C_3 -symmetric BZ of the BLG/hBN moiré.

We thus believe that BLG supermoiré systems host exciting states that are absent in SLG-based systems and are thus it is stimulating to probe them in their own right.

We have added the following discussion in page 3 of the revised manuscript:

Recent tunneling spectroscopy studies establish that unlike single-layer graphene (SLG)/hBN single moiré, the BLG/hBN single moiré is gapless [8]. It is instructive to study if the same distinction persists between the hBN/SLG/hBN and hBN/BLG/hBN supermoiré. As we show in this letter, this is not the case – while BLG single moiré is gapless, BLG supermoiré has multiple gaps even at zero displacement field. We also demonstrate that the BLG supermoiré is different from its single-layer counterpart in other critical aspects – for example, in the symmetry of the moiré Brillouin zone, which has direct consequences for the anomalous Hall effect [9] and electron-electron scattering [1, 2]. Additionally, the ability to electrically control the layer and valley degrees of freedom in BLG promises exotic phases that are absent in its single-layer counterpart, e.g. *electric field switchable* Chern insulators [3].

Comment 3: Can you comment on why the electron gap is non-zero, while in BG on hBN (without encapsulation) the gap is expected to be zero due to band overlap [see Nano Lett. 2018, 18, 12, 7732–7741 (2018)]. Do you have a band structure for zero-degree aligned hBN-encapsulated BG to visualize this behavior and see how the encapsulation opens up this gap? You can also do it at finite angles if you find a small enough commensurate cell for the double moire system but the zero-degree alignment should already provide some information.

Response: We thank the referee for pointing this out. As correctly pointed out by the referee, the electron gap is expected to be zero for bilayer graphene on hBN due to band overlap (without encapsulation). While checking the origin of this gap in our calculations, we discovered an artifact arising due to a subtle bug in the Hamiltonian for the top moiré. We have fixed the bug and have benchmarked the band structure with the published band structures (Fig. 2(a,b) of this document) reported in ref [10] by turning off the top and bottom moiré potentials, one at a time.

Based on our revised calculations, we find that the supermoiré potential lifts the band overlap, leading to the appearance of multiple gaps on the hole side (negative energy side of the electronic spectrum). On the other hand, as the Reviewer suggested, the electron side (positive energy) indeed remains gapless. This is in conformity with previous calculations [11]. The modified results are consistent with the argument of electron-hole asymmetric splitting, because of which there should be no electron gap due to band overlap. This is apparent in the plots of DOS vs number density n for all angles, $\theta_t = 0.41^\circ, 0.44^\circ, 0.47^\circ, 0.52^\circ$ (Fig. S8 Supplementary Information). In Fig. 2(c) of this document, we provide the band structure for zero-degree aligned hBN-encapsulated BLG to visualize this behaviour.

We have updated the revised manuscript with the results of the revised calculation. We find a significant improvement in the quantitative agreements of our theoretical results with the experimental data. As shown in the revised manuscript, this also led

FIG. 2. **Band structures for zero-degree aligned BLG/hBN and hBN/BLG/hBN** (a) and (b) shows the band structures for BLG/hBN at 0° arising from the bottom and top moiré potentials respectively. The slight differences in the bands are because of different types of stackings of hBN on BLG (type-1 and type-2) as addressed in ref [10]. (c) Calculated band structure for zero-degree aligned hBN-encapsulated BLG – one can see that there is no gap on the electron side. In (a-c), the red line are for bands arising from K -valley and the blue lines are for bands arising from K' -valley.

to a better physical interpretation of our data. Additionally, in the revised manuscript, we now identify the location of the van Hove singularities.

Again, we are extremely grateful to the reviewer for raising this issue, which helped us correct a bug in our calculations.

We have added the following statements in the revised manuscript:

On page 3: Recent tunneling spectroscopy studies establish that unlike single-layer graphene (SLG)/hBN single moiré, the BLG/hBN single moiré is gapless [8]. It is instructive to study if the same distinction persists between the hBN/SLG/hBN and hBN/BLG/hBN supermoiré. As we show in this letter, this is not the case – while BLG single moiré is gapless, BLG supermoiré has multiple gaps even at zero displacement field.

On page 7: Note also that there are no gaps at positive energies with the exception of the supermoiré gap at $n_s = 0.39 \times 10^{16} \text{ m}^{-2}$ ($\bar{1}, \bar{1}, 1, 1$). The reason why the supermoiré gap survives the band overlaps (that quenches all other gaps at positive energies) is at present unclear.

Comment 4: Do you know if the hBN layers have their B and N atoms on top of each other respectively for 0-degree alignments or the B atom is on top of the N atom and the N atom is on top of the B atom, corresponding to a 60-degree rotation of the top layer?

Response: We thank the Reviewer for raising this important point. We do not know directly from experiments whether the hBN layers are aligned at 0° or 60° . However, as the results of our theoretical calculations by the continuum model depend critically on the nature of stacking, we can rule out the stacking orders that *do not* match our experimental observations.

Recall that there are five possible stacking configurations of the hBN/BLG/hBN heterostructure (Fig. 3 of this document) [12]. We find that Bragg gaps are generated at the primary and the secondary Dirac points exclusively for the **AB1** stacking. Despite

FIG. 3. Schematic of the five stacking arrangements of hBN layers the B and N atoms are shown in orange and blue, respectively. The image is based on ref. [12].

the experimental uncertainty regarding the exact stacking configuration, the consistency between the AB1 stacking calculations and experimental results suggests that the stacking of B and N is of the AB1 type.

We have added a new section in Supplementary Materials (section S3. Determining Stacking type) where we discuss this issue.

Comment 5: Presumably being in the commensurate limit, how does this affect your estimates of the twist angles based on the moiré length?

Response: In the commensurate limit, the dependence of the twist angle on the moiré length is given by the following relation:

$$\lambda = \frac{(1 + \epsilon)a}{[\epsilon^2 + 2(1 + \epsilon)(1 - \cos(\theta))]^{1/2}} \quad (1)$$

Here $a = 0.246$ nm is the lattice constant of graphene, ϵ is the lattice mismatch between the hBN and graphene, θ is the relative rotational angle between the two lattices, and λ is the wavelength of commensurate supercell. Thus, the only factors that affect the estimate of twist angles based on the moiré length in the commensurate limit are the lattice constants of graphene and hBN. This point has been discussed in Supplementary materials section S2.

Comment 6: Similarly, how does potential strain affect your angle estimates?

Response: As Eqn. 1 shows, potential strain in the system can substantially affect the twist angle estimates. This was also shown by recent studies [13, 14]. In our calculations, we took the rigid continuum model approach with a lattice constant of graphene $a = 0.246$ nm and a lattice constant of hBN to be 0.2504 nm. This leads to a strain difference of $\epsilon = 0.018$. We also used other values of the strain parameter in the commonly used range $0.0165 \leq \epsilon \leq 0.0185$. The theoretical results generated using $\epsilon = 0.018$ match the best with our measured experimental results.

We have added the following statement on page 6 in the revised manuscript:

Note that in our theoretical calculations, we assumed the lattice constant of graphene and hBN to be 0.246 nm and 0.2504 nm, respectively (corresponding to a strain, $\epsilon = 0.018$). We also considered other values of the strain parameter in the commonly used range $0.0165 \leq \epsilon \leq 0.0185$. The theoretical values generated with $\epsilon = 0.018$ match best with our measured experimental results.

Comment: Small comment 1: f is at some point referred to as a frequency, later on as a periodicity. Can you remove this ambiguity?

Response: We thank the Reviewer for pointing this out; this ambiguity has been removed. In the revised manuscript, f is referred to as the frequency.

Comment: Small comment 2: Can you define f_s the first time you introduce it in the text (maybe I missed it)?

Response: We thank the referee for pointing this out. In the revised manuscript, $f_s = 4$ T has been defined as the supermoiré frequency of Brown-Zak oscillations.

We have added the following statement on page 5 in the revised manuscript:

The Brown Zak frequency $f_s = 4$ T yields $n_s = 0.39 \times 10^{16} \text{ m}^{-2}$ – this number density corresponds to a real-space wavelength of $\lambda_s = 34.6$ nm which is the size of the supermoiré unit cell in our heterostructure (Supplementary Materials S2). We thus identify f_s to be the supermoiré Brown Zak frequency.

We sincerely thank the referee for their very careful review and insightful comments that helped us improve the manuscript.

REPORT OF REVIEWER 2 NCOMMS-23-19425-T

Comment 1: The authors study the moire superlattice effect in the electronic structure in bilayer graphene doubly aligned with top and bottom hBN layers. By the electronic transport and magneto-transport measurements, they observed a number of Bragg gaps opening in the low-energy spectrum, which are confirmed to quantitatively agree with the theoretical calculation using a continuum model. As a major finding of the paper, they identified the topological numbers and associated quasi-Brillouin zones for major gaps using the theoretical approach proposed in Ref. [13]. This is the first work that experimentally validates the topological gap labeling in quasi-crystalline systems, and I find it is potentially worth publication in Nature Communications. I request the authors to address the following problems, concerning the correspondence between the experimental results and the theoretical calculation.

Response: We thank the Reviewer for the valuable feedback on our work and finding our work potentially worthy of publishing in Nature Communications.

Comment 2: In Fig.2f, the Hall carrier density n_H diverges at five points. I understand each of them corresponds to a band gap, and the sign of n_H indicates the carrier type (electron/holes). When going down from the Dirac point, for instance, n_H diverges to the negative infinity at $n = 2.2 \times 10^{16}/\text{m}^2$ and then it comes back from positive infinity, suggesting that we have electron bands and hole bands above and below the gap, respectively, as naturally expected. However, the behaviour becomes opposite in the next diverging point ($n = 2.4 \times 10^{16}$); i.e., we have holes above and electrons below the gap, and it's alternating for the 3rd, 4th and 5th. This is an interesting part, but it's hard to imagine from the band calculation because we naively expect a band bottom above the gap and a band top below the gap. Can the band calculation simulate this behaviour? We also should have the van-Hove singularity somewhere in the middle of the band, at which the carrier type changes while not diverging. Also, the four-number gap labeling is missing for the fourth diverging point ($n=-2.5$), while the divergence behavior is quite pronounced. Any possibility to have a big gap between two main gaps? Based on these considerations, I remain somewhat unconvinced about the agreement between the experimental results and the theoretical calculations.

Response: We thank the referee for raising this important question. We realize that the explanation provided in the original manuscript was confusing and unclear. As correctly pointed out by the reviewer, $n_H (R_{xy}(B))$ can change sign both at the band gaps and at van-Hove singularities. A careful analysis of the measured n_H sign change, calculated value of DOS, and the measured d^2G_{xx}/dn^2 data reveals that at $n = -2.05 \times 10^{16}/\text{m}^2$, $-2.6 \times 10^{16}/\text{m}^2$ the system has van-Hove singularities, while at $n = -2.4 \times 10^{16}/\text{m}^2$, $-2.8 \times 10^{16}/\text{m}^2$ band gaps open. The band gaps marked in the figures of the revised manuscript are now consistent with the expectation of the carrier-type changes across the gap.

In Fig.S9 of the Supplementary materials, we plot the unfolded bands along a path for different strengths of moiré potentials. For a potential strength of 0.2, we see the electron and hole bands above and below the gaps, as expected and as pointed out by the reviewer. However, for the full strength of moiré potential, the spectral weights get redistributed along a path, making it harder to track the bands and pinpoint the sign changes close to the gap. An accurate mapping of the carrier densities above and below a gap requires the calculation of the Hall coefficient, which is outside the scope of the present study. While theoretically, it is possible to have a big gap between two main gaps – we do not see this in our experimental data.

We have changed the discussion on page 6 of the revised manuscript to the following:

The zeroes (and several prominent non-zero dips) in the calculated DOS are reflected in the experimental data as a discontinuity in the $n_H - n$ plot. Recall that in a multi-carrier type system and for small B , a change in sign of R_{xy} (or a corresponding divergence in n_H) can either indicate a bandgap or a van Hove singularity [15, 16]. The sign of n_H on either side of a band gap

reflects the local band curvature (and hence, the carrier type). Thus, for instance, with $E_F > 0$, one can have both positive and negative n_H ; a positive (negative) value of n_H implies an electron-like (a hole-like) band (we take the electronic charge to be e). A band gap can be said to exist at a certain number density if the following three conditions are simultaneously met: (1) the DOS in Fig.3(a) goes to zero, (2) n_H in Fig.3(c) changes sign, and (3) there is a local maximum in the d^2G_{xx}/dn^2 (minima in G_{xx}) data in Fig.3(d). Using this criterion, we identify the principal gaps at $n_b = -2.36 \times 10^{16} \text{ m}^{-2}$, $n_t = -2.80 \times 10^{16} \text{ m}^{-2}$, and $n_{CNP} = 0 \text{ m}^{-2}$ as Bragg gaps with quantum numbers $(0, \bar{1}, 0, 0)$, $(\bar{1}, 0, 0, 0)$, $(0, 0, 0, 0)$, and respectively. We also identify several higher-order Bragg gaps, for example, at $n = -3.3 \times 10^{16} \text{ m}^{-2}$ $(2, 2, \bar{1}, \bar{4})$ and $-4.8 \times 10^{16} \text{ m}^{-2}$ $(\bar{4}, \bar{1}, 0, 3)$. We mark all the Bragg gaps with solid gray lines in Fig.3(a-f).

There are certain number densities for example, at $\pm 0.39 \times 10^{16} \text{ m}^{-2}$ $(\bar{1}, \bar{1}, 1, 1)$ and $-6.6 \times 10^{16} \text{ m}^{-2}$ $(\bar{1}, 2, \bar{3}, \bar{1})$, (marked by dotted blue lines in Fig.3(a-d)) where the DOS goes to zero and the G_{xx} has a minima, however n_H does not reach zero. We tentatively identify them as narrow Bragg gaps that are masked by thermal/impurity broadening. Note also that there are no gaps at positive energies with the exception of the supermoiré gap at $n_s = 0.39 \times 10^{16} \text{ m}^{-2}$ $(\bar{1}, \bar{1}, 1, 1)$. The reason why the supermoiré gap survives the band overlaps (that quenches all other gaps at positive energies) is at present unclear.

Additionally, there are features at which n_H changes sign accompanied by minima in d^2G_{xx}/dn^2 (maxima in G_{xx}) and a peak in DOS – we identify these to be due to van Hove singularities. Two of these (at $n = -2.05 \times 10^{16} \text{ m}^{-2}$ and $n = -2.6 \times 10^{16} \text{ m}^{-2}$) have been marked with purple dotted lines in Fig.3(e-f). Note that, in addition to the ones marked, the calculated DOS plotted in Fig.3(a) shows several dips at which the measured longitudinal and Hall resistances are featureless. We find that at these points, either the DOS is finite with no band gap, or the calculated band gaps Δ are substantially smaller than 1 meV (e.g. at $n = -2.18 \times 10^{16} \text{ m}^{-2}$, $\Delta = 0.74 \text{ meV}$) and hence not resolvable in our electrical transport measurements.

Comment 3: The correlation between the text and figures is not very cohesive and it is sometimes hard to relate. For instance, the n_b, n_t , etc should be indicated in Fig. 1 and Fig. 2. Also, it would be convenient if the four numbers are also indicated in fig. 2e and 2f.

Response: We realize that the presentation of the data was not optimal in the original manuscript. We have addressed this issue in the revised manuscript. As per the reviewer's suggestion, changes have been made to the figures to make the text and figures cohesive. We have added the labels n_b, n_t , etc, in the figures of the revised manuscript. We have also included the four numbers in Fig. 2(e) and 2(f) (Fig. 3(e) and (f) of the revised manuscript).

Comment 4: What do gray solid/dashed/dotted lines represent in Fig. 2?

Response: The lines in Fig.2 (Fig.3 in the revised manuscript) have been redrawn to reduce clutter and make the presentation more accessible.

In the revised figure, the solid grey lines represent the locations of the band gaps. These have been identified using the following criterion: At these number densities, the following conditions must be satisfied simultaneously (1) the theoretical density of states (DOS) has a significant dip, (2) the measured n_H changes sign, and (3) the second derivatives of the conductance has a local maximum.

The dotted blue line represents the location of band gaps for which the measured n_H do not change sign.

The dotted purple lines mark the carrier densities for which DOS has a local maximum, the second derivative of conductance shows minima (maxima in conductance), and n_H changes its sign. We identify these to be van-Hove singularities.

We have added the following statements in the caption of the old Fig.2 (Fig.3 in the revised manuscript):

In (a-d), the solid gray lines mark the values of n at which the Bragg gaps open with $\text{DOS}=0$, $R_{xy} = 0$ and $d^2G_{xx}(B=0)/dn^2$ having a maxima; the dotted blue lines mark the values of n at which the Bragg gaps open with $\text{DOS}=0$ and $d^2G_{xx}(B=0)/dn^2$ having a maxima, but R_{xy} does not reach zero. (e-f) Zoomed-in plots of DOS and n_H versus n in a narrow range on the hole-side. The solid gray lines indicate the Bragg gaps, while the dotted purple lines indicate the locations of the van Hove singularities.

Comment 5: The Brown-Zak frequency f_s seems mentioning 4T but not defined in the text.

Response: We thank the referee for pointing out this oversight on our part. In the revised manuscript, $f_s = 4 \text{ T}$ has been defined as the supermoiré frequency of Brown-Zak oscillations.

We have added the following statements on page 5 of the revised manuscript:

The Brown Zak frequency $f_s = 4 \text{ T}$ yields $n_s = 0.39 \times 10^{16} \text{ m}^{-2}$ – this number density corresponds to a real-space wavelength of $\lambda_s = 34.6 \text{ nm}$ which is the size of the supermoiré unit cell in our heterostructure (Supplementary Materials S2). We thus identify f_s to be the supermoiré Brown Zak frequency.

Comment 6: Is θ_b fixed to 0.026 deg in model p1 to p4 in Sec. S4?

Response: Yes, in section S4 (Section S5 of revised supplementary material) of supplementary material, θ_b is fixed to 0.026 deg. Only θ_t has been varied from 0.4 degrees to 0.52 degrees in order to trace the path of gaps, which eventually leads to the determination of unique zone quantum numbers.

We thank the referee once again for the careful and constructive review that helped us prepare a much-improved manuscript.

REPORT OF REVIEWER 3 NCOMMS-23-19425-T

Comment 1: The study of moiré superlattices has been emerging in the past few years, yet there are still many open questions in this field. One of the perplexing problems is the moiré quasiperiodicity, or a system with two moiré wavelengths that interfere. This manuscript studies a bilayer graphene double aligned with top and bottom hBN, and observes supermoiré patterns which can be explained by a simple picture of linear combination of periodic areas. This analysis bridges a gap in this field, which I believe will be of general interest. However, I think the data quality is below standard and some of the data interpretation is not convincing enough. Especially, the raw experimental data is too scarce that cannot fully support the argument. Therefore, I don't think it meets the criteria of Nature Communications. Below are my comments.

Response: We greatly appreciate the valuable feedback provided by the Reviewer. We hope that the additional data presented in the revised manuscript and the revised analysis will convince the Reviewer that the study is suitable for publication in Nature Communications.

Comment 2: I don't see the superlattice peak of supermoiré. Do the authors observe it in the resistance measurement? If not, what might be the reason that they don't see it?

Response: We appreciate this observation of the reviewer. We do not see the superlattice peak in the resistance versus carrier density response; this is in agreement with the previous study on the bilayer graphene supermoiré system (see ref. [17]). There can be two possible reasons for not observing the supermoiré peak in resistance:

1. We estimate the magnitude of the supermoiré gap to be $\Delta_s \approx 0.63$ meV – it is unlikely that such a small gap will give rise to a substantial resistance peak.
2. From the measured Brown-Zak oscillations, we estimate the carrier density at which the superlattice gap should open to be $n_s = 0.39 \times 10^{16} \text{ m}^{-2}$. We do not see a clear peak at this carrier density, but we do find that the FWHM of the charge neutrality point is greatly increased for the supermoiré as compared to that of the single moiré device (see Fig.S3 of Supplementary materials). We believe that the supermoiré peak, if it appeared in resistance measurements, would be subsumed by the peak of the primary Dirac point.

We have added the preceding discussion in the revised Supplementary Materials as a new section.

Comment 3: The authors should present the raw data of the Landau fan diagram for readers to get more information of the device. The Brown-Zak oscillation data is too vague and it's hard to tell more than one oscillation period by observing the 2D map. At least the authors should present a linecut at specific doping before presenting its the Fourier spectrum in Fig.1f.

Response: We agree with the Reviewer's critique. To address them, we have added:

1. the raw data of the Landau fan diagram as Fig. 1(e) of the revised manuscript.
2. a new section (Section S9. Landau fan diagram) in the Supplementary Materials discussing the Landau fan diagram data..
3. a new figure (Fig. 2 of the revised manuscript) where we show the line-cuts of the Brown-Zak oscillations for carrier density $n = 3.3 \times 10^{16} \text{ m}^{-2}$ and its Fourier spectra.

We also have added the following statements on page 4 of the revised manuscript:

Fig.1(e) shows the 2D map of $G_{xx}(n, B)$ at $D = 0$ V/nm in the $n - B$ plane – one finds Landau fans emerging from the charge neutrality point (CNP) and from the secondary Dirac points n_b and n_t with Landau filling $\nu = \pm 4m$ ($m \in \text{integer}$) (Supplementary Materials S9). The faint horizontal streaks in the plot are the Brown Zak oscillations originating from the recurring Bloch states in the superlattice [18, 19]. These features get accentuated at high temperatures, where thermal smearing

diminishes the effect of Landau quantization on the magnetotransport. This is seen clearly from Fig.1(a) which presents the magnetoconductance $\Delta G_{xx}(B)$ plotted in the $n - 1/B$ plane; the data were measured at 100 K. The Fourier transform of a representative data measured at $n = 3.3 \times 10^{16} \text{ m}^{-2}$ (Fig.2(b)) yields multiple frequencies $f = 24.5 \text{ T}, 29 \text{ T}, \text{ and } 4 \text{ T}$ (Fig.2(c)).

Comment 4: In Fig. 1d, there are a few solid lines and dashed lines for guiding, but these lines are too thick that I don't even see the features beneath them. Also, what are the dashed lines referring to?

Response: We agree with the Reviewer's observations. As per the Reviewer's suggestions, to make the underlying features in Fig. 1(d) clearly visible, the lines have been replaced by colored arrows outside the plot.

A line plot of the resistance data versus carrier density, measured for $D = 0 \text{ V/nm}$ is shown in Fig. 1(c). The various features of interest in the plot have been marked with colored arrows. The black dashed line indicates the charge neutrality point. The magenta and green arrows indicate the secondary Dirac points emerging from top and bottom moiré respectively, with carrier density (moiré wavelength) $n_t = 2.80 \times 10^{16} \text{ m}^{-2}$ ($\lambda_t = 12.84 \text{ nm}$) and $n_b = 2.36 \times 10^{16} \text{ m}^{-2}$ ($\lambda_b = 13.97 \text{ nm}$). Colored arrows in Fig. 1(d) have the same interpretation as in Fig. 1(c). Gray Dashed line in Fig 1(d) represented the higher order Bragg features, to reduce the clutter these lines have been removed from the revised Fig1(d). We believe that the revised representation makes the data easier to understand.

The caption of Fig. 1 has been revised to read:

Characteristics of the double moiré device. (a) Schematic of doubly aligned BLG with top and bottom hBN. The black and the white hexagons mark the primary moiré and supermoiré plaquettes, respectively. (b) An optical image of the device (before adding the top gate) labeled with the measurement configuration. Top inset: Schematic of the layer-stacking, with the direction of increasing displacement field D marked. (c) Plot of the longitudinal resistance $R_{xx}(B = 0)$ as a function of n . The black dashed line marks the charge neutrality point. Magenta and dark green lines indicate the secondary Dirac points emerging from top and bottom moiré respectively, with carrier density (moiré wavelength) $n_t = \pm 2.80 \times 10^{16} \text{ m}^{-2}$ ($\lambda_t = 12.84 \text{ nm}$) and $n_b = \pm 2.36 \times 10^{16} \text{ m}^{-2}$ ($\lambda_b = 13.97 \text{ nm}$), respectively. (d) Map of R_{xx} as a function of the back gate voltage, V_{bg} and top gate voltage V_{tg} . The color of the arrows have the same interpretation as in (c). (e) Landau-fan diagram $G_{xx}(n, B)$ showing the emergence of Landau levels from the primary Dirac point and the two secondary Dirac points. The cyan horizontal arrows on the right of the plot mark the weak Brown Zak features. The measurements were done at $T = 2 \text{ K}$.

Comment 5: In Fig.2, the authors compare the Hall measurement with calculated DOS. However, there are too many lines and it's difficult to locate where the authors are referring to as "additional peaks" or "main features". Both the notations in figures and in the manuscript should be clearer. There should be a baseline of "0" drawn in the R_{xy} and n_H figures for easier discussion.

Response: We acknowledge the Reviewer's observation that the terms 'additional peaks' and 'main features' were misleading. To rectify this, we have made significant improvements to enhance the clarity of the manuscript. Specifically, we have introduced and clearly defined the terms 'Bragg gap' and 'van Hove singularity', which accurately convey the nature of the observed phenomenon.

The '0' baseline has been added in the R_{xy} and n_H plots.

Comment 6: Some of the features in DOS, R_{xy} and n_H do match, but there are many features in the DOS. For example, the following line pointed by red doesn't necessarily correspond to a gap feature in DOS. Also, why are there some other prominent dips in DOS not shown as peaks in R_{xx} or divergence in R_{xy} , such as the one pointed by blue?

Response: We thank the referee for raising this important question of some feature mismatch. While addressing this question, we discovered a minor bug in the Hamiltonian of the top moiré. We fixed the bug and benchmarked the band structure with the band structures (See Fig. 2(a,b) of this document) reported in [10] by turning off the top and bottom moiré potentials, one at a time.

We also acknowledge that the presentation of the data in the original manuscript was confusing. As rightly pointed out by the Reviewer, not all sign changes in n_H correspond to dips in DOS. These are, in fact, the van-Hove singularities which we had missed marking distinct from the band gaps in the original manuscript. In the revised figure, the solid grey lines represent the locations of the band gaps. These have been identified using the following criterion: At these number densities, the following conditions must be satisfied simultaneously (1) the theoretical density of states (DOS) has a significant dip, (2) the measured n_H changes sign, and (3) the second derivatives of the conductance has a local maximum. The dotted blue line represents the location of band gaps for which the measured n_H do not change sign. The dotted purple lines mark the carrier densities for which DOS has a local maximum, the second derivative of conductance shows minima (maxima in conductance), and n_H changes its sign. We identify these to be van-Hove singularities.

At the point marked by the Reviewer by blue arrow (and in other similar points where the DOS has a significant dip but the R_{xy} and R_{xx} and mostly featureless). There can be two reasons for this:

FIG. 4. First figure accompanying the third Reviewer’s queries.

1. The size of these band gaps is very small. For example, at $n = -3.66 \times 10^{16} / \text{m}^2$, the gap is of the order of 1 meV – this is of the order of impurity broadening of the levels (as detected through quantum oscillation measurements) and hence is not resolved in our transport measurements.
2. Ours is a non-interacting calculation. Some higher-order gaps emerging in non-interacting DOS may disappear in the interacting picture as observed in the experiments leading to a featureless R_{xx} or R_{xy} at those number densities.

Motivated by the comments 5 and 6 of the Reviewer, we have changed the discussion on page 6 of the revised manuscript to the following:

The zeroes (and several prominent non-zero dips) in the calculated DOS are reflected in the experimental data as a discontinuity in the $n_H - n$ plot. Recall that in a multi-carrier type system and for small B , a change in sign of R_{xy} (or a corresponding divergence in n_H) can either indicate a bandgap or a van Hove singularity [15, 16]. The sign of n_H on either side of a band gap reflects the local band curvature (and hence, the carrier type). Thus, for instance, with $E_F > 0$, one can have both positive and negative n_H ; a positive (negative) value of n_H implies an electron-like (a hole-like) band (we take the electronic charge to be e). A band gap can be said to exist at a certain number density if the following three conditions are simultaneously met: (1) the DOS in Fig.3(a) goes to zero, (2) n_H in Fig.3(c) changes sign, and (3) there is a local maximum in the d^2G_{xx}/dn^2 (minima in G_{xx}) data in Fig.3(d). Using this criterion, we identify the principal gaps at $n_b = -2.36 \times 10^{16} \text{ m}^{-2}$, $n_t = -2.80 \times 10^{16} \text{ m}^{-2}$, and $n_{CNP} = 0 \text{ m}^{-2}$ as Bragg gaps with quantum numbers $(0, \bar{1}, 0, 0)$, $(\bar{1}, 0, 0, 0)$, $(0, 0, 0, 0)$, and respectively. We also identify several higher-order Bragg gaps, for example, at $n = -3.3 \times 10^{16} \text{ m}^{-2}$ $(2, 2, \bar{1}, \bar{4})$ and $-4.8 \times 10^{16} \text{ m}^{-2}$ $(\bar{4}, \bar{1}, 0, 3)$. We mark all the Bragg gaps with solid gray lines in Fig.3(a-f).

There are certain number densities for example, at $\pm 0.39 \times 10^{16} \text{ m}^{-2}$ $(\bar{1}, \bar{1}, 1, 1)$ and $-6.6 \times 10^{16} \text{ m}^{-2}$ $(\bar{1}, 2, \bar{3}, \bar{1})$, (marked by dotted blue lines in Fig.3(a-d)) where the DOS goes to zero and the G_{xx} has a minima, however n_H does not reach zero. We tentatively identify them as narrow Bragg gaps that are masked by thermal/impurity broadening. Note also that there are no gaps at positive energies with the exception of the supermoiré gap at $n_s = 0.39 \times 10^{16} \text{ m}^{-2}$ $(\bar{1}, \bar{1}, 1, 1)$. The reason why the supermoiré gap survives the band overlaps (that quenches all other gaps at positive energies) is at present unclear.

Additionally, there are features at which n_H changes sign accompanied by minima in d^2G_{xx}/dn^2 (maxima in G_{xx}) and a peak in DOS – we identify these to be due to van Hove singularities. Two of these (at $n = -2.05 \times 10^{16} \text{ m}^{-2}$ and $n = -2.6 \times 10^{16} \text{ m}^{-2}$) have been marked with purple dotted lines in Fig.3(e-f). Note that, in addition to the ones marked, the calculated DOS plotted in Fig.3(a) shows several dips at which the measured longitudinal and Hall resistances are featureless. We find that at these points, either the DOS is finite with no band gap, or the calculated band gaps Δ are substantially smaller than 1 meV (e.g. at $n = -2.18 \times 10^{16} \text{ m}^{-2}$, $\Delta = 0.74 \text{ meV}$) and hence not resolvable in our electrical transport measurements.

Comment 7: The authors mention that the system is a ferroelectric phase, so does it have hysteresis when scanning either gate backward and forward? Does it having “layer-specific anomalous screening” behavior as discussed in Nature 588, 71–76 (2020)? Which gate sweeping direction did the authors choose for their dual-gate scan? Although the ferroelectricity and supermoiré pattern can be independent, the authors should present the details of the raw experimental data instead of claiming it showing ferroelectricity without illustrating any details.

Response: We apologize for that confusing statement in the original manuscript. We did not probe for ferroelectricity in

this manuscript. We only meant to convey for the sake of completeness that recent studies [5, 6] claimed to have observed a ferroelectric phase in this part of the phase diagram.

To reiterate, we do not claim that the system is ferroelectric nor is this the focus of this study.

In the revised manuscript, we have removed this statement about the ferroelectric phase.

Comment 8: Is there any specific reason for using bilayer graphene for the study? Do the authors expect anything different if they use a monolayer graphene (e.g. Science Advances 5, eaay8897 (2019) used a monolayer for similar studies)?

Response: We agree with the Reviewer that the motivation behind studying bilayer-based supermoiré was not brought out clearly in our original manuscript. As the Reviewer pointed out correctly, single-layer and bilayer graphene-based supermoiré systems have certain similarities, principal among them are: (1) for a given twist angle, the moiré wavelength and the number densities at which Bragg gaps open are the same for both systems and (2) the concept of higher order fractal gap applies to both systems. However, as we list below, there are significant differences between these two material platforms.

1. As we show in this letter, the shapes of the Brillouin zone are very different for the two systems (Fig. 1 of this document). For the bilayer graphene supermoiré, the Brillouin zone is a trigonally distorted hexagon with C_3 symmetry, which is a consequence of intrinsic trigonal warping in pristine bilayer graphene. On the other hand, for single-layer graphene supermoiré, the Brillouin zone is hexagonal in shape. (Figs. S10 and S11 of the Supplementary Materials).
2. This difference directly affects the umklapp scattering in these systems. The umklapp threshold carrier density is constant for single-layer graphene moiré [1], and it becomes a non-monotonic function of twist angle for bilayer graphene moiré system [2], which is a consequence of the trigonally warped Brillouin zone.
3. Unlike single-layer graphene supermoiré, bilayer graphene supermoiré systems are predicted to host electric field switchable Chern insulators [3].
4. Experimental tunneling spectroscopy studies suggest that in an SLG single moiré, a bandgap appears at the secondary Dirac points, while the BLG single moiré is gapless at all number densities [4]. Thus it is instructive to compare the bands of supermoiré in SLG and BLG to check if the same distinction persists. Our studies establish that this is not the case – supermoiré opens a gap at the secondary Dirac point in BLG (in contrast to single moiré in BLG that fails to open a bandgap at the secondary Dirac point).

These distinctions were our motivation behind probing the supermoiré state in a bilayer graphene-based system. We have added the following discussion on page 3 in the revised manuscript:

Recent tunneling spectroscopy studies establish that unlike single-layer graphene (SLG)/hBN single moiré, the BLG/hBN single moiré is gapless [8]. It is instructive to study if the same distinction persists between the hBN/SLG/hBN and hBN/BLG/hBN supermoiré. As we show in this letter, this is not the case – while BLG single moiré is gapless, BLG supermoiré has multiple gaps even at zero displacement field. We also demonstrate that the BLG supermoiré is different from its single-layer counterpart in other critical aspects – for example, in the symmetry of the moiré Brillouin zone, which has direct consequences for the anomalous Hall effect [9] and electron-electron scattering [1, 2]. Additionally, the ability to electrically control the layer and valley degrees of freedom in BLG promises exotic phases that are absent in its single-layer counterpart, e.g. *electric field switchable* Chern insulators [3].

Comment 9: In the supplementary, the authors reproduce the data of Science Advances 5, eaay8897 (2019), but one of the data points should be $\pm 4.1 \times 10^{12} \text{ cm}^{-2}$ instead of $\pm 4.35 \times 10^{12} \text{ cm}^{-2}$ in the original paper. I think this is why the line for guidance doesn't even correspond to a peak in R_{xx} in the supplementary (pointed by red).

Response: We thank the Reviewer for pointing this out. We indeed find that the R_{xx} shows a peak, and theoretically calculated DOS also shows zero at the carrier density $n = -4.1 \times 10^{16} \text{ m}^{-2}$, corresponding to which Bragg indices are $(1, 0, \bar{1}, \bar{2})$. Again, we thank the Reviewer for bringing this to our notice. We have revised Fig. S12 accordingly and modified the discussion following it in the revised Supplementary Materials:

Peaks that went unexplained in the original publication at $n = -3.10 \times 10^{16} \text{ m}^{-2}$ and $n = -4.1 \times 10^{16} \text{ m}^{-2}$ can now be understood to arise due to the noticeable zeros in the calculated density of states and correspond to the Bragg indices $(1, \bar{1}, 1, \bar{2})$ and $(1, 0, \bar{1}, \bar{2})$ respectively.

We appreciate the detailed and thorough review that helped us improve our manuscript significantly.

FIG. 5. Second figure accompanying the third Reviewer's queries.

LIST OF SIGNIFICANT CHANGES MADE IN THE FIGURES OF THE MAIN MANUSCRIPT:

1. Figure 1 of the main manuscript has been modified to include a line plot of the measured zero magnetic field longitudinal resistance and the Landau fan diagram. The figures have been replotted again for clarity keeping the Reviewer's suggestions in mind..
2. Figure 2 of the main manuscript is a new figure. It shows (a) the Brown Zak oscillations in the $n - 1/B$ plane, (b) a line plot of the oscillations at a particular number density, and (c) its Fourier spectrum.

LIST OF SIGNIFICANT CHANGES MADE IN THE TEXT OF THE MAIN MANUSCRIPT:

1. On page 3, the following discussion presents the motivation for studying bilayer supermoiré structures:

Recent tunneling spectroscopy studies establish that unlike single-layer graphene (SLG)/hBN single moiré, the BLG/hBN single moiré is gapless [8]. It is instructive to study if the same distinction persists between the hBN/SLG/hBN and hBN/BLG/hBN supermoiré. As we show in this letter, this is not the case – while BLG single moiré is gapless, BLG supermoiré has multiple gaps even at zero displacement field. We also demonstrate that the BLG supermoiré is different from its single-layer counterpart in other critical aspects – for example, in the symmetry of the moiré Brillouin zone, which has direct consequences for the anomalous Hall effect [9] and electron-electron scattering [1, 2]. Additionally, the ability to electrically control the layer and valley degrees of freedom in BLG promises exotic phases that are absent in its single-layer counterpart, e.g. *electric field switchable* Chern insulators [3].

2. On page 3, the statement regarding ferroelectricity has been removed.

3. On page 4, we provide details of the R_{xx} measured at $B = 0$:

The device is in a dual-gated field-effect transistor architecture, allowing independent control on the charge carrier density n and displacement field D via $n = [(C_{tg}V_{tg} + C_{bg}V_{bg})/e + n_0]$ and $D = [(C_{bg}V_{bg} - C_{tg}V_{tg})/e + D_0]$ across the device. Here C_{bg} (C_{tg}) is the back-gate (top-gate) capacitance, and V_{bg} (V_{tg}) is the back-gate (top-gate) voltage. The values of C_{tg} and C_{bg} are determined from quantum Hall measurements. n_0 and D_0 are the residual charge carrier number density and displacement field due to channel impurities, respectively. A plot of the longitudinal resistance R_{xx} measured at $D = 0$ and zero magnetic field is shown in Fig.1(c). The appearance of split moiré resistance peaks at $n_b = \pm 2.36 \times 10^{16} \text{ m}^{-2}$ and $n_t = \pm 2.80 \times 10^{16} \text{ m}^{-2}$ indicates the alignment of the BLG with both the bottom and top hBN layers. Their presence is also apparent in the 2D map of R_{xx} in the $V_{bg} - V_{tg}$ plane (Fig.1 (d)).

4. On page 4, we provide details of the Landau fan diagram and Brown Zak oscillations:

Fig.1(e) shows the 2D map of $G_{xx}(n, B)$ at $D = 0 \text{ V/nm}$ in the $n - B$ plane – one finds Landau fans emerging from the charge neutrality point (CNP) and from the secondary Dirac points n_b and n_t with Landau filling $\nu = \pm 4m$ ($m \in \text{integer}$) (Supplementary Materials S9). The faint horizontal streaks in the plot are the Brown Zak oscillations originating from the recurring Bloch states in the superlattice [18, 19]. These features get accentuated at high temperatures, where thermal smearing diminishes the effect of Landau quantization on the magnetotransport. This is seen clearly from Fig.1(a) which presents the magnetoconductance $\Delta G_{xx}(B)$ plotted in the $n - 1/B$ plane; the data were measured at 100 K. The Fourier transform of a representative data measured at $n = 3.3 \times 10^{16} \text{ m}^{-2}$ (Fig.2(b)) yields multiple frequencies $f = 24.5 \text{ T}$, 29 T, and 4 T (Fig.2(c)).

The Brown Zak frequency $f_s = 4 \text{ T}$ yields $n_s = 0.39 \times 10^{16} \text{ m}^{-2}$ – this number density corresponds to a real-space wavelength of $\lambda_s = 34.6 \text{ nm}$ which is the size of the supermoiré unit cell in our heterostructure (Supplementary Materials S2). We thus identify f_s to be the supermoiré Brown Zak frequency.

5. On page 6 we discuss the possible effect of strain on the calculated moiré angles:

Note that in our theoretical calculations, we used the lattice constant of graphene and hBN to be 0.246 nm and 0.2504 nm, respectively (corresponding to a strain, $\epsilon = 0.018$). We also considered other values of the strain parameter in the commonly used range $0.0165 \leq \epsilon \leq 0.0185$. The theoretical values generated with $\epsilon = 0.018$ match best with our measured experimental results.

6. On pages 6 and 7, we have modified the discussion of the observed Bragg peaks. We also identified the van Hove singularities:

Using the above formalism, the band gaps corresponding to the densities n_b , n_t , and n_s are identified to be Bragg gaps with Bragg indices $(0, 1, 0, 0)$, $(1, 0, 0, 0)$, and $(1, 1, \bar{1}, \bar{1})$ respectively (see Supplementary Materials S5). We obtain the positions of additional Bragg gaps by comparing calculated DOS (Fig.3(a)) with the experimentally determined transverse resistance $R_{xy}(B)$ and the extracted Hall carrier density $n_H = B/(eR_{xy})$ measured in the presence of a small, non-quantizing magnetic field $B = 0.7$ T (Fig.3(b-c)).

The zeroes (and several prominent non-zero dips) in the calculated DOS are reflected in the experimental data as a discontinuity in the $n_H - n$ plot. Recall that in a multi-carrier type system and for small B , a change in sign of R_{xy} (or a corresponding divergence in n_H) can either indicate a bandgap or a van Hove singularity [15, 16]. The sign of n_H on either side of a band gap reflects the local band curvature (and hence, the carrier type). Thus, for instance, with $E_F > 0$, one can have both positive and negative n_H ; a positive (negative) value of n_H implies an electron-like (a hole-like) band (we take the electronic charge to be e). A band gap can be said to exist at a certain number density if the following three conditions are simultaneously met: (1) the DOS in Fig.3(a) goes to zero, (2) n_H in Fig.3(c) changes sign, and (3) there is a local maximum in the d^2G_{xx}/dn^2 (minima in G_{xx}) data in Fig.3(d). Using this criterion, we identify the principal gaps at $n_b = -2.36 \times 10^{16} \text{ m}^{-2}$, $n_t = -2.80 \times 10^{16} \text{ m}^{-2}$, and $n_{CNP} = 0 \text{ m}^{-2}$ as Bragg gaps with quantum numbers $(0, \bar{1}, 0, 0)$, $(\bar{1}, 0, 0, 0)$, $(0, 0, 0, 0)$, and respectively. We also identify several higher-order Bragg gaps, for example, at $n = -3.3 \times 10^{16} \text{ m}^{-2}$ $(2, 2, \bar{1}, \bar{4})$ and $-4.8 \times 10^{16} \text{ m}^{-2}$ $(\bar{4}, \bar{1}, 0, 3)$. We mark all the Bragg gaps with solid gray lines in Fig.3(a-f).

There are certain number densities for example, at $\pm 0.39 \times 10^{16} \text{ m}^{-2}$ $(\bar{1}, \bar{1}, 1, 1)$ and $-6.6 \times 10^{16} \text{ m}^{-2}$ $(\bar{1}, 2, \bar{3}, \bar{1})$, (marked by dotted blue lines in Fig.3(a-d)) where the DOS goes to zero and the G_{xx} has a minima, however n_H does not reach zero. We tentatively identify them as narrow Bragg gaps that are masked by thermal/impurity broadening. Note also that there are no gaps at positive energies with the exception of the supermoiré gap at $n_s = 0.39 \times 10^{16} \text{ m}^{-2}$ $(\bar{1}, \bar{1}, 1, 1)$. The reason why the supermoiré gap survives the band overlaps (that quenches all other gaps at positive energies) is at present unclear.

Additionally, there are features at which n_H changes sign accompanied by minima in d^2G_{xx}/dn^2 (maxima in G_{xx}) and a peak in DOS – we identify these to be due to van Hove singularities. Two of these (at $n = -2.05 \times 10^{16} \text{ m}^{-2}$ and $n = -2.6 \times 10^{16} \text{ m}^{-2}$) have been marked with purple dotted lines in Fig.3(e-f). Note that, in addition to the ones marked, the calculated DOS plotted in Fig.3(a) shows several dips at which the measured longitudinal and Hall resistances are featureless. We find that at these points, either the DOS is finite with no band gap, or the calculated band gaps Δ are substantially smaller than 1 meV (e.g. at $n = -2.18 \times 10^{16} \text{ m}^{-2}$, $\Delta = 0.74$ meV) and hence not resolvable in our electrical transport measurements.

LIST OF SIGNIFICANT CHANGES MADE IN THE SUPPLEMENTARY MATERIALS:

1. We added a new section discussing the possible stacking order (S3. Determining the stacking type)
2. We added a new section discussing the Landau fan data (S9. Landau fan diagram).
3. We added a new section discussing the possible reasons for the absence of the supermoiré peak in resistance measurements (S10. Absence of supermoiré peak in resistance measurements).
4. We have corrected our analysis of the data from Ref. [13].

-
- [1] J. R. Wallbank, R. Krishna Kumar, M. Holwill, Z. Wang, G. H. Auton, J. Birkbeck, A. Mishchenko, L. A. Ponomarenko, K. Watanabe, T. Taniguchi, K. S. Novoselov, I. L. Aleiner, A. K. Geim, and V. I. Fal'ko, Excess resistivity in graphene superlattices caused by umklapp electron-electron scattering, *Nature Physics* **15**, 32 (2019).
 - [2] C. Mouldale and V. Fal'ko, Umklapp electron-electron scattering in bilayer graphene moiré superlattice, *Phys. Rev. B* **107**, 144111 (2023).
 - [3] M. M. A. Ezzi, J. Hu, Ariando, F. Guinea, and S. Adam, Topological flat bands in graphene super-moiré lattices (2023), [arXiv:2306.10116 \[cond-mat.mes-hall\]](https://arxiv.org/abs/2306.10116).
 - [4] H. Kim, N. Leconte, B. L. Chittari, K. Watanabe, T. Taniguchi, A. H. MacDonald, J. Jung, and S. Jung, Accurate gap determination in monolayer and bilayer graphene/h-bn moiré superlattices, *Nano Letters* **18**, 7732 (2018).
 - [5] R. Niu, Z. Li, X. Han, Z. Qu, D. Ding, Z. Wang, Q. Liu, T. Liu, C. Han, K. Watanabe, T. Taniguchi, M. Wu, Q. Ren, X. Wang, J. Hong, J. Mao, Z. Han, K. Liu, Z. Gan, and J. Lu, Giant ferroelectric polarization in a bilayer graphene heterostructure, *Nature Communications* **13**, 6241 (2022).

- [6] Z. Zheng, Q. Ma, Z. Bi, S. de la Barrera, M.-H. Liu, N. Mao, Y. Zhang, N. Kiper, K. Watanabe, T. Taniguchi, J. Kong, W. A. Tisdale, R. Ashoori, N. Gedik, L. Fu, S.-Y. Xu, and P. Jarillo-Herrero, Unconventional ferroelectricity in moiré heterostructures, *Nature* **588**, 71 (2020).
- [7] Z. Zhu, S. Carr, Q. Ma, and E. Kaxiras, Electric field tunable layer polarization in graphene/boron-nitride twisted quadrilayer superlattices, *Phys. Rev. B* **106**, 205134 (2022).
- [8] H. Kim, N. Leconte, B. L. Chittari, K. Watanabe, T. Taniguchi, A. H. MacDonald, J. Jung, and S. Jung, Accurate gap determination in monolayer and bilayer graphene/h-bn moiré superlattices, *Nano Letters* **18**, 7732 (2018), PMID: 30457338, <https://doi.org/10.1021/acs.nanolett.8b03423>.
- [9] A. Khalifa, G. Murthy, and R. K. Kaul, Lattice model for the quantum anomalous hall effect in moiré graphene, *Phys. Rev. B* **107**, 235137 (2023).
- [10] P. Moon and M. Koshino, Electronic properties of graphene/hexagonal-boron-nitride moiré superlattice, *Phys. Rev. B* **90**, 155406 (2014).
- [11] H. Oka and M. Koshino, Fractal energy gaps and topological invariants in hbn/graphene/hbn double moiré systems, *Phys. Rev. B* **104**, 035306 (2021).
- [12] R. Smeyers, M. V. Milošević, and L. Covaci, Strong gate-tunability of flat bands in bilayer graphene due to moiré encapsulation between hbn monolayers, *Nanoscale* **15**, 4561 (2023).
- [13] Z. Wang, Y. B. Wang, J. Yin, E. Tóvári, Y. Yang, L. Lin, M. Holwill, J. Birkbeck, D. J. Perello, S. Xu, J. Zultak, R. V. Gorbachev, A. V. Kretinin, T. Taniguchi, K. Watanabe, S. V. Morozov, M. Anđelković, S. P. Milovanović, L. Covaci, F. M. Peeters, A. Mishchenko, A. K. Geim, K. S. Novoselov, V. I. Falko, A. Knothe, and C. R. Woods, Composite super-moiré lattices in double-aligned graphene heterostructures, *Science Advances* **5**, eaay8897 (2019), <https://www.science.org/doi/pdf/10.1126/sciadv.aay8897>.
- [14] J. Jung, A. M. DaSilva, A. H. MacDonald, and S. Adam, Origin of band gaps in graphene on hexagonal boron nitride, *Nature Communications* **6**, 6308 (2015).
- [15] P. He, G. K. W. Koon, H. Isobe, J. Y. Tan, J. Hu, A. H. C. Neto, L. Fu, and H. Yang, Graphene moiré superlattices with giant quantum nonlinearity of chiral Bloch electrons, *Nature Nanotechnology* **17**, 378 (2022).
- [16] S. Wu, L. Wang, Y. Lai, W.-Y. Shan, G. Aivazian, X. Zhang, T. Taniguchi, K. Watanabe, D. Xiao, C. Dean, J. Hone, Z. Li, and X. Xu, Multiple hot-carrier collection in photo-excited graphene moiré superlattices, *Science Advances* **2**, e1600002 (2016), <https://www.science.org/doi/pdf/10.1126/sciadv.1600002>.
- [17] M. Kuiri, S. K. Srivastava, S. Ray, K. Watanabe, T. Taniguchi, T. Das, and A. Das, Enhanced electron-phonon coupling in doubly aligned hexagonal boron nitride bilayer graphene heterostructure, *Phys. Rev. B* **103**, 115419 (2021).
- [18] J. Zak, Magnetic translation group, *Phys. Rev.* **134**, A1602 (1964).
- [19] E. Brown, Bloch electrons in a uniform magnetic field, *Phys. Rev.* **133**, A1038 (1964).

REVIEWER COMMENTS

Reviewer #1 (Remarks to the Author):

The authors have properly addressed my concerns. I do not have further reservations against publication in Nature Communications.

Reviewer #2 (Remarks to the Author):

The revised manuscript demonstrates improved readability and more coherent notations/definitions. The issues regarding the identification of mini-gaps and van Hove singularities have been corrected. Additionally, the newly-appended Landau fan diagram [Fig. 1e] lends support to the completeness of the experimental data.

However, I cannot recommend the paper for publication in Nature Communications for the following reasons:

On careful examination of the data, I find that the gap labeling in the paper appears reliable only for $(-1,0,0,0)$ and $(0,-1,0,0)$, which represent the first-order gaps of the individual moire patterns. At the points claimed to be higher-order gaps, such as $(2,2,-1,-4)$ and $(-4,-1,0,3)$, R_{xy} vanishes (n_H diverges), while corresponding features in R_{xx} [Figs. 1c, 1d, 1e] (also in d^2G_{xx}/dn^2 in Fig. 3d) are hardly visible. Moreover, there are no notable structures in the band calculation presented in Fig. S6. These higher-order labelings appear rather arbitrary and lack conclusive evidence.

In contrast, the gap labeling presented in Fig. S12, based on the reproduced data from Science Advances 5, eaay8897 (2019) (SLG + double hBN system), appears quite convincing. The band gaps in the calculated DOS (including higher-order gaps) align perfectly with the peaks observed in R_{xx} . It would be desirable to achieve a similar level of matching in the present bilayer case.

To summarize, I find that the current data only provides reliable gap labeling for the first-order gaps, which are also observed in the single moire system. Therefore, I conclude that this is not sufficient for publication in Nature Communications, as the novelty of the paper should lie in the identification of topological numbers in the double moire system.

Reviewer #3 (Remarks to the Author):

The authors have improved their manuscript significantly, both in data presentation and interpretation. I think this work provides a systematic way to characterize supermoire systems which is beneficial to the community and now has potential for publication in Nature Communications. However, there are still some concerns that I hope the authors can address.

1. In terms of the novelty of this work, which was pointed out by myself and also other reviewers, the authors mainly argue about the difference between bilayer and monolayer graphene. I wonder how much their theoretical investigations is different from previous studies, e.g. Science Advances 5, eaay8897 (2019). Has it been carried out in any other previous studies? The authors should comment about the novelty of their work also in this aspect.

2. The authors argue that the BLG single moire is gapless experimentally, which I have concerns about. There is only one experimental paper the authors cited on this, which is not convincing enough. Also, how do the authors define gapped/gapless? In the current manuscript, the authors have two devices, one with single moire and the other with double moire. Do these two devices agree with the authors' argument about gapped vs gapless?

3. The authors removed some guiding lines in the dual-gate map and made the map clearer. However, it is clear to me that there is one additional peak on the low density side on electron side, and one on the high density side on the hole side. They are visible in the dual-gate map and the 1D linecut. They used to have some guiding lines on them in the previous manuscript, but now there is no comment at all. Where do they come from?

4. For claiming the two resistance peaks are from top and bottom moires, respectively, I think the

authors should mention about their control single moire device in the main manuscript. Otherwise, I think some audience would naturally assign them to be sample twist angle inhomogeneity.

The Reviewers' comments are in bold, and our responses, inserted after each specific point, are in regular font. This is followed by a list of significant changes made in the manuscript.

REPORT OF REVIEWER 1 NCOMMS-23-19425A

Comment 1: The authors have properly addressed my concerns. I do not have further reservations against publication in Nature Communications.

Response: We thank the Reviewer for their thorough and timely evaluation of our manuscript and their recommendation to publish it in nature communication.

REPORT OF REVIEWER 2 NCOMMS-23-19425A

Comment 1: The revised manuscript demonstrates improved readability and more coherent notations/definitions. The issues regarding the identification of mini-gaps and van Hove singularities have been corrected. Additionally, the newly appended Landau fan diagram [Fig. 1e] lends support to the completeness of the experimental data.

Response: We thank the Reviewer for the feedback and constructive suggestions. We are pleased to hear that the revision has enhanced the readability and coherence of the manuscript.

Comment 2: However, I cannot recommend the paper for publication in Nature Communications for the following reasons. On careful examination of the data, I find that the gap labeling in the paper appears reliable only for $(-1, 0, 0, 0)$ and $(0, -1, 0, 0)$, which represent the first-order gaps of the individual moire patterns. At the points claimed to be higher-order gaps, such as $(2, 2, -1, -4)$ and $(-4, -1, 0, 3)$, R_{xy} vanishes (n_H diverges), while corresponding features in R_{xx} [Figs. 1c, 1d, 1e] (also in d^2G_{xx}/dn^2 in Fig. 3d) are hardly visible. Moreover, there are no notable structures in the band calculation presented in Fig. S6. These higher-order labelings appear rather arbitrary and lack conclusive evidence.

Response: We thank the Reviewer for finding the gap labeling reliable for the first order gaps $(\bar{1}, 0, 0, 0)$ and $(0, \bar{1}, 0, 0)$.

As correctly pointed out by the Reviewer, at the points claimed to be higher-order gaps, such as $(2, 2, \bar{1}, \bar{4})$ and $(\bar{4}, \bar{1}, 0, 3)$, Hall carrier density n_H diverges. We agree that the data presented in the original manuscript on R_{xx} [Figs. 1c, 1d, 1e] (also in d^2G_{xx}/dn^2 in Fig. 3d) were less than convincing. To make the presence of these higher-order gaps more persuasive, we present the results of new measurements in the revised manuscript. The features at carrier densities $n = -3.3 \times 10^{16} \text{ m}^{-2}$ and $n = -4.8 \times 10^{16} \text{ m}^{-2}$ corresponding higher-order gaps with quantum numbers $(2, 2, \bar{1}, \bar{4})$ and $(\bar{4}, \bar{1}, 0, 3)$ respectively are clearly visible in Fig. 1(c) of the revised manuscript. The corresponding d^2G_{xx}/dn^2 2D plot in Fig.3(d) has also been updated with higher-resolution data, showing clearly the feature at these carrier densities, which persist over a range of electric fields. The position of the Bragg gaps in carrier density labeled as $(2, 2, \bar{1}, \bar{4})$ and $(\bar{4}, \bar{1}, 0, 3)$ over a range of electric fields has also been added in Fig.3(g). We believe this new data will convince the Reviewer that we have clear signatures of higher-order Bragg gaps at these number densities.

As pointed out by the Reviewer, the higher order features labeled as $(2, 2, \bar{1}, \bar{4})$ and $(\bar{4}, \bar{1}, 0, 3)$ do not show a notable gap in the density of states (DOS) versus carrier density (n) for $\theta_t = 0.44^\circ$, but are prominent dips (Fig.S8). Motivated by this concern, we realized there can be a minute angle-inhomogeneity in the system. As discussed in the revised Supplementary information file, we estimate this uncertainty in θ to be $\delta\theta = 0.03^\circ$. Thus, the twist angles lie in the range $\theta_b = 0.03^\circ \pm 0.03^\circ$ and $\theta_t = 0.44^\circ \pm 0.03^\circ$. The observed features will be a convolution of the DOS versus n for the angles around $\theta_b = 0.03^\circ \pm 0.03^\circ$ and $\theta_t = 0.44^\circ \pm 0.03^\circ$.

Allowing for this slight variation in the angle by $\delta\theta = 0.03^\circ$, we observe that several higher-order features become more discernible (Fig. S8 of Supplementary information). For example, for $\theta_t = 0.47^\circ$, the feature labeled as $(\bar{1}, 2, \bar{3}, \bar{1})$ becomes a gap (with the DOS going to zero), and the dip corresponding to $(2, 2, \bar{1}, \bar{4})$ becomes much more prominent. Feature labeled as $(\bar{4}, \bar{1}, 0, 3)$ appears as a dip consistently in DOS for all the four angles ($\theta_t = 0.40^\circ$, $\theta_t = 0.44^\circ$, $\theta_t = 0.47^\circ$, $\theta_t = 0.52^\circ$) (See Fig.S8). There can be several angles within this range with larger commensurate cells required for the calculation, where this dip in DOS may turn into a gap.

We believe the new data and analysis will convince the Reviewer that the gap identification is grounded on solid reasoning.

We have added the following statements in the revised manuscript:

- In the main text: Fig. 1(c) and Fig. [3d, 3g] have been updated with the results of new measurements.
- On page no 5 of the main text: We have added the twist angle uncertainty, $\theta_b = 0.03^\circ \pm 0.03^\circ$ and $\theta_t = 0.44^\circ \pm 0.03^\circ$.
- In Supplementary Materials Section S2, we have added: In the device, the calculated impurity carrier density is ($\delta n_i \approx 7 \times 10^{14} \text{ m}^{-2}$), which gives an uncertainty in twist angle estimation of $\delta\theta \approx 0.03^\circ$.

Comment 3: In contrast, the gap labeling presented in Fig. S12, based on the reproduced data from Science Advances 5, eaay8897 (2019) (SLG + double hBN system), appears quite convincing. The band gaps in the calculated DOS (including higher-order gaps) align perfectly with the peaks observed in R_{xx} . It would be desirable to achieve a similar level of matching in the present bilayer case. To summarize, I find that the current data only provides reliable gap labeling for the first-order gaps, which are also observed in the single-moire system. Therefore, I conclude that this is not sufficient for publication in Nature Communications, as the novelty of the paper should lie in the identification of topological numbers in the double moire system.

Response: We agree with the Reviewer's observation that our analysis of the data on SLG/hBN super moiré matches exceptionally well with the experimental data reproduced from the Science

Advances 5, eaay8897 (2019) [1]. The match between theory and experiment in the present study on BLG/hBN supermoiré has improved considerably with our new measurements and analysis. As we show below, any lingering discrepancy is attributable to the distinctions intrinsic to the system.

Our calculations of the DOS for both the SLG/hBN double moiré system and BLG/hBN double moiré system at the same twist angles ($\theta_b = 0.03^\circ$ and $\theta_t = 0.44^\circ$) using the continuum model are benchmarked with Ref. [2, 3]. It shows that most of the prominent dips observed in the BLG double moiré system convert to sizable band gaps in the SLG double moiré system (Supplementary Materials, section S11). We also find that the size of the band gaps E_g is consistently larger for SLG double moiré, in comparison to BLG double moiré (see table I), making their identification from measurements and DOS analysis easier. We are grateful to the Reviewer for raising this issue, which led us to this important new finding that forms a significant distinction between these two systems with important implications for possible applications.

n ($\times 10^{16} \text{m}^{-2}$)	E_g for BLG (meV)	E_g for SLG (meV)	quantum numbers
-0.39	0.63	1.06	$\bar{1}\bar{1}11$
-2.36	7.70	8.85	$0\bar{1}00$
-2.80	0.25	4.17	$\bar{1}000$
-4.8	-	1.98	$\bar{4}\bar{1}03$

TABLE I. Gap size comparison between BLG and SLG double moiré for twist angles $\theta_b = 0.03^\circ$ and $\theta_t = 0.44^\circ$.

We do not as yet have a clear physical understanding of this difference. One possible intuitive reasoning is that the bilayer graphene supermoiré features are affected by interlayer hopping between the two graphene layers. The moiré potential dies exponentially with the distance in real space, making the effect of the lower (top) hBN on the top (lower) graphene layer of the BLG weaker. On the other hand, for the SLG/hBN supermoiré, there is no such effect giving rise to better features of higher-order Bragg gaps.

We hope that the improved data and analysis will persuade the Reviewer that the present study indeed identifies topological numbers in double moiré systems in both BLG/hBN (based on current data) and in SLG/hBN (based on data from Ref. [1]) and to recommend the publication of the manuscript in Nature Communications.

We have added a new section, S11, in the revised manuscript supplementary materials:

- S11 Comparing hBN double moiré system of SLG and BLG

Fig.S14 shows the calculated DOS for both the SLG/hBN double moiré system and BLG/hBN double moiré system at the same twist angles ($\theta_b = 0.03^\circ$ and $\theta_t = 0.44^\circ$) using the continuum model. It shows that most of the prominent dips observed in the BLG double moiré system convert to sizable band gaps in the SLG double moiré system. We also find that the size of the band gaps E_g is consistently larger for SLG double moiré, in comparison to BLG double moiré (see table I). The exact reasoning behind this is unclear and requires further studies.

We are grateful for the critical review by the Reviewer, which helped improve the quality of the manuscript tremendously.

REPORT OF REVIEWER 3 NCOMMS-23-19425A

Comment 1: The authors have improved their manuscript significantly, both in data presentation and interpretation. I think this work provides a systematic way to characterize supermoire systems which is beneficial to the community and now has potential for publication in Nature Communications.

Response: We thank the Reviewer for their thorough evaluation of our manuscript, as well as for their comment ‘is beneficial to the community and now has potential for publication in Nature Communications.’

Comment 2: In terms of the novelty of this work, which was pointed out by myself and also other reviewers, the authors mainly argue about the difference between bilayer and monolayer graphene. I wonder how much their theoretical investigations is different from previous studies, e.g. Science Advances 5, eaay8897 (2019). Has it been carried out in any other previous studies? The authors should also comment about their work’s novelty in this aspect.

Response: To address this quest, we first describe the difference between the theoretical models used in the previous studies and the present work. The study on SLG/hBN supermoiré (Science Advances 5, eaay8897 (2019)) [1] uses the Hamiltonian described within the symmetry-based approach [4], which uses the moiré potential parameters determined from the study of transverse magnetic focusing in graphene/hBN superlattices [5]. The peaks observed in the experimental R_{xx} vs n plot are validated by matching with the Bragg vectors, $G_m^b - G_k^t$.

However, our study uses the continuum model developed by P Moon et al. [6]. The parameters of this model are extracted from the tight-binding calculations for the systems with different angles. We extended this to write the model Hamiltonian for the hBN/BLG/hBN supermoiré system. This model explains the Bragg gaps corresponding to all possible combinations, $pG_1^b + qG_2^b + rG_1^t + sG_2^t$.

We validate the experimental findings of R_{xx} and d^2G_{xx}/dn^2 versus n by side-by-side mapping with the calculated DOS, along with the characterization of gaps with the quantum numbers and quasi-Brillouin zones. To our knowledge, this is the first analysis of higher-order gaps in the BLG double-moiré system.

In the revised manuscript, we present a comparison of the results of this calculation between the SLG/hBN supermoiré system (as studied in Science Advances 5, eaay8897 (2019)) [1] and our BLG/hBN supermoiré system. We find essential distinctions between these two systems regarding the positions and magnitudes of the higher-order Bragg gaps. We are not aware of any previous study that compares these two systems.

We are unaware of any previous study comparing the continuum model-based DOS and the experimental transport data for such supermoiré systems.

We have added the following statements in the revised manuscript:

- On page no 3 in the main text we have added: This model explains the Bragg gaps corresponding to the linear combinations of moiré reciprocal lattice vectors, $pG_1^b + qG_2^b + rG_1^t + sG_2^t$. Additionally, our analysis explains several unexplained experimental features in graphene/hBN supermoiré systems reported in recent publications [1] (Supplementary Material S8), which were previously studied based on symmetry-based approach [4].
- On pages no 3-4, in the main text, we have added: We demonstrate that the BLG supermoiré is different from its single-layer counterpart in several critical aspects – for example, in the symmetry of the moiré Brillouin zone, which has direct consequences for the anomalous Hall effect [7] and electron-electron scattering [8, 9], in terms of the positions and magnitudes of the higher-order Bragg gaps (Supplementary Material S11).

Comment 3: The authors argue that the BLG single moire is gapless experimentally, which I have concerns about. There is only one experimental paper the authors cited on this, which is not convincing enough. Also, how do the authors define gapped/gapless? In the current manuscript, the authors have two devices, one with single moire and the other with double moire. Do these two devices agree with the authors' argument about gapped vs gapless?.

Response: As per the Reviewer's suggestion, we performed temperature-dependent resistance

measurements on the BLG single moiré device. We observed activated behaviour at primary moiré gaps on the hole side with the band gap of ≈ 2.4 meV. Therefore, we have removed the following statements from the revised manuscript.:

- On Page no 3 of the main text, we have removed: Recent tunneling spectroscopy studies establish that unlike single-layer graphene (SLG)/hBN single moiré, the BLG/hBN single moiré is gapless [10]. It is instructive to study if the same distinction persists between the hBN/SLG/hBN and hBN/BLG/hBN supermoiré. As we show in this letter, this is not the case – while BLG single moiré is gapless, BLG supermoiré has multiple gaps even at zero displacement field.

We thank the Reviewer for raising this point that motivated us to measure the gap in the single-moiré device.

Comment 4: The authors removed some guiding lines in the dual-gate map and made the map clearer. However, it is clear to me that there is one additional peak on the low density side on electron side, and one on the high density side on the hole side. They are visible in the dual-gate map and the 1D linecut. They used to have some guiding lines on them in the previous manuscript, but now there is no comment at all. Where do they come from?

Response: We thank the referee for pointing this out. The additional peak that the Reviewer has pointed out on the hole side is at carrier density $n = -3.3 \times 10^{16} \text{ m}^{-2}$ corresponds to the higher order Bragg gaps with quantum number $(2, 2, \bar{1}, \bar{4})$. We have marked and identified this peak in the revised manuscript Fig. 1(c-d).

As the Reviewer also pointed out, there is one additional peak on the electron side at $n = 2 \times 10^{16} \text{ m}^{-2}$. However, we find neither a dip in the calculated DOS nor an activated behaviour of R_{xx} in our measurements at this carrier density. Given this, the origin of this peak is not clear to us.

We have added the following changes to the revised manuscript:

- In the revised manuscript Fig. 1(c-d) of the main text: We have marked and identified the higher order Bragg gaps with quantum number $(2, 2, \bar{1}, \bar{4})$ as a peak in R_{xx} at carrier density $n = -3.3 \times 10^{16} \text{ m}^{-2}$.

Comment 5: For claiming the two resistance peaks are from top and bottom moires, respec-

tively, I think the authors should mention about their control single moire device in the main manuscript. Otherwise, I think some audience would naturally assign them to be sample twist angle inhomogeneity.

Response: We thank the Reviewer for this excellent suggestion.

We have added the following statements in the revised manuscript:

- On page no. 5 of the main text, we have added: To verify that the split peaks at n_b and n_t are not artifacts due to large angle-inhomogeneity in the device, we repeated the measurements on a control device (labeled D_{single}) where only the top-hBN forms a moiré with the BLG (Supplementary Materials, S2). A single-layer WSe₂ was interposed between half of the BLG and the lower hBN to achieve this. The $n - R_{xx}$ plot of this single-moiré device had a single secondary peak at $n = n_t$ (Supplementary Materials, Fig.S3).

We thank the Reviewer for the constructive critique that helped us improve the manuscript.

LIST OF SIGNIFICANT CHANGES MADE IN THE FIGURES OF THE MAIN MANUSCRIPT:

1. Fig. 1(c) and Fig. [3d, 3g] have been updated with the results of new measurements to show clearly the features of higher-order Bragg gaps.

LIST OF SIGNIFICANT CHANGES MADE IN THE TEXT OF THE MAIN MANUSCRIPT:

1. On page 5 of the main text: We have added the twist angle uncertainty. $\theta_b = 0.03^\circ \pm 0.03^\circ$ and $\theta_t = 0.44^\circ \pm 0.03^\circ$.
2. To address the novelty of work. On page no 3 in the main text, we have added: This model explains the Bragg gaps corresponding to the linear combinations of moiré reciprocal lattice vectors, $pG_1^b + qG_2^b + rG_1^t + sG_2^t$. Additionally, our analysis explains several unexplained experimental features in graphene/hBN supermoiré systems reported in recent publications [1] (Supplementary Material S8), which were previously studied based on symmetry-based approach [4].
3. On pages 3-4 we have also added the following statement: We demonstrate that the BLG supermoiré is different from its single-layer counterpart in several critical aspects – for example, in the symmetry of the moiré Brillouin zone, which has direct consequences for the anomalous Hall effect [7] and electron-electron scattering [8, 9], in terms of the positions and magnitudes of the higher-order Bragg gaps (Supplementary Material S11).
4. On page no 3, we have removed the following statement: Recent tunneling spectroscopy studies establish that unlike single-layer graphene (SLG)/hBN single moiré, the BLG/hBN single moiré is gapless [10]. It is instructive to study if the same distinction persists between the hBN/SLG/hBN and hBN/BLG/hBN supermoiré. As we show in this letter, this is not the case – while BLG single moiré is gapless, BLG supermoiré has multiple gaps even at zero displacement field.
5. On page 5 of the main text, we have added the following statement: To verify that the split peaks at n_b and n_t are not artifacts due to large angle-inhomogeneity in the device, we repeated the measurements on a control device (labeled D_{single}) where only the top-hBN forms a moiré with the BLG (Supplementary Materials, S2). To achieve this, a single-layer

WSe₂ was interposed between half of the BLG and the lower hBN. The $n - R_{xx}$ plot of this single-moiré device had a single secondary peak at $n = n_t$ (Supplementary Materials, Fig.S3).

LIST OF SIGNIFICANT CHANGES MADE IN THE SUPPLEMENTARY MATERIALS:

1. In Supplementary Materials Section S2, we have added the following statement: **In the device, the calculated impurity carrier density is ($\delta n_i \approx 7 \times 10^{14} \text{ m}^{-2}$), which gives an uncertainty in twist angle estimation of $\delta\theta \approx 0.03^\circ$.**
2. We have added a new section S11 in Supplementary Materials, comparing the hBN double moiré system of SLG and BLG.

-
- [1] Z. Wang, Y. B. Wang, J. Yin, E. Tóvári, Y. Yang, L. Lin, M. Holwill, J. Birkbeck, D. J. Perello, S. Xu, J. Zultak, R. V. Gorbachev, A. V. Kretinin, T. Taniguchi, K. Watanabe, S. V. Morozov, M. Anđelković, S. P. Milovanović, L. Covaci, F. M. Peeters, A. Mishchenko, A. K. Geim, K. S. Novoselov, V. I. Fal'ko, A. Knothe, and C. R. Woods, Composite supermoiré; lattices in double-aligned graphene heterostructures, *Science Advances* **5**, eaay8897 (2019), <https://www.science.org/doi/pdf/10.1126/sciadv.aay8897>.
 - [2] H. Oka and M. Koshino, Fractal energy gaps and topological invariants in hbn/graphene/hbn double moiré systems, *Phys. Rev. B* **104**, 035306 (2021).
 - [3] P. Moon and M. Koshino, Electronic properties of graphene/hexagonal-boron-nitride moiré superlattice, *Physical Review B* **90**, 155406 (2014).
 - [4] J. R. Wallbank, A. A. Patel, M. Mucha-Kruczyński, A. K. Geim, and V. I. Fal'ko, Generic miniband structure of graphene on a hexagonal substrate, *Phys. Rev. B* **87**, 245408 (2013).
 - [5] M. Lee, J. R. Wallbank, P. Gallagher, K. Watanabe, T. Taniguchi, V. I. Fal'ko, and D. Goldhaber-Gordon, Ballistic miniband conduction in a graphene superlattice, *Science* **353**, 1526 (2016), <https://www.science.org/doi/pdf/10.1126/science.aaf1095>.
 - [6] P. Moon and M. Koshino, Electronic properties of graphene/hexagonal-boron-nitride moiré superlattice, *Phys. Rev. B* **90**, 155406 (2014).

- [7] A. Khalifa, G. Murthy, and R. K. Kaul, Lattice model for the quantum anomalous hall effect in moiré graphene, *Phys. Rev. B* **107**, 235137 (2023).
- [8] J. R. Wallbank, R. Krishna Kumar, M. Holwill, Z. Wang, G. H. Auton, J. Birkbeck, A. Mishchenko, L. A. Ponomarenko, K. Watanabe, T. Taniguchi, K. S. Novoselov, I. L. Aleiner, A. K. Geim, and V. I. Falko, Excess resistivity in graphene superlattices caused by umklapp electron-electron scattering, *Nature Physics* **15**, 32 (2019).
- [9] C. Mouldale and V. Fal'ko, Umklapp electron-electron scattering in bilayer graphene moiré superlattice, *Phys. Rev. B* **107**, 144111 (2023).
- [10] H. Kim, N. Leconte, B. L. Chittari, K. Watanabe, T. Taniguchi, A. H. MacDonald, J. Jung, and S. Jung, Accurate gap determination in monolayer and bilayer graphene/h-bn moiré superlattices, *Nano Letters* **18**, 7732 (2018), pMID: 30457338, <https://doi.org/10.1021/acs.nanolett.8b03423>.

REVIEWERS' COMMENTS

Reviewer #2 (Remarks to the Author):

The new data in the revised manuscript (Fig. 1c and Fig. 3d) exhibits the features of the higher-order gaps much more clearly. Regarding the correspondence to the theoretical calculations, I see in Fig. 8 that the DOS dips for higher-order gaps $(2,2,-1,-4)$, $(-4,-1,0,3)$, $(-1,2,3,1)$ appear robust against twist angle fluctuations. The paper also confirms that the same analysis applies to the previous experimental results [Science Advances 5, eaay8897 (2019)](SLG + double hBN system), which is valuable in itself. Overall, I found that the manuscript demonstrates gap labeling based on the quasi-Brillouin zone area in real systems for the first time. I conclude that it is suitable for publication in Nature Communications.

I request the authors to address the following minor problems.

--- The relation between the newly added arrows in Fig. 1c (at $n = -3.3 \times 10^{16} \text{ m}^{-2}$ and $n = -4.8 \times 10^{16} \text{ m}^{-2}$) and Fig. 3 is unclear (i.e., which part in Fig.3 are these arrows corresponding to?). Maybe some symbols are needed?

--- It would be useful to present the band structures responsible for the higher-order dips of $(2,2,-1,-4)$, $(-4,-1,0,3)$, $(-1,2,3,1)$. The band diagram in Fig.S6 could cover these regions, with the corresponding energies clearly marked. I understand they are not gaps, but they may be related to some sparsely-distributed band regions.
===

Reviewer #3 (Remarks to the Author):

The authors have addressed my questions and I would like to recommend publication in Nature Communications.

The Reviewer's comments are in bold, and our responses, inserted after each specific point, are in regular font. This is followed by a list of significant changes made in the manuscript.

REPORT OF REVIEWER 2 NCOMMS-23-19425B

Comment 1: The new data in the revised manuscript (Fig. 1c and Fig. 3d) exhibits the features of the higher-order gaps much more clearly. Regarding the correspondence to the theoretical calculations, I see in Fig. 8 that the DOS dips for higher-order gaps (2,2,-1,-4), (-4,-1,0,3), (-1,2,3,1) appear robust against twist angle fluctuations. The paper also confirms that the same analysis applies to the previous experimental results [Science Advances 5, eaay8897 (2019)](SLG + double hBN system), which is valuable in itself. Overall, I found that the manuscript demonstrates gap labeling based on the quasi-Brillouin zone area in real systems for the first time. I conclude that it is suitable for publication in Nature Communications.

Response: We thank the Reviewer for their thorough and timely evaluation of our manuscript and their recommendation to publish it in nature communication.

Comment 2: The relation between the newly added arrows in Fig. 1c (at $n = -3.3 \times 10^{16} \text{ m}^{-2}$ and $n = -4.8 \times 10^{16} \text{ m}^{-2}$) and Fig. 3 is unclear (i.e., which part in Fig.3 are these arrows corresponding to?). Maybe some symbols are needed?

Response: We agree with the reviewer that the visibility of the arrows was not optimum. In the revised manuscript, for better visibility of the Bragg gaps, we have modified the color plot of Fig.3d. The Bragg gap at carrier density $n = -3.3 \times 10^{16} \text{ m}^{-2}$ and $n = -4.8 \times 10^{16} \text{ m}^{-2}$ with corresponding quantum numbers (2,2,-1,-4), (-4,-1,0,3), respectively has been marked with gray arrow lines.

Comment 3: It would be useful to present the band structures responsible for the higher-order dips of (2,2,-1,-4), (-4,-1,0,3), (-1,2,3,1). The band diagram in Fig.S6 could cover these regions, with the corresponding energies clearly marked. I understand they are not gaps, but they may be related to some sparsely-distributed band regions.

Response: In Supplementary Figure 6, We have extended the data till -150 meV, which covers the band structure for the higher-order dips of quantum number (2,2,-1,-4), (-4,-1,0,3) and (-1,2,-3,-1). The energy regions have been marked for these quantum numbers. As also correctly pointed out by the reviewer, the bands in these energy regions are indeed very sparse and have been shaded for better visibility. Again, we thank the reviewer for the valuable suggestions.

REPORT OF REVIEWER 3 NCOMMS-23-19425B

Comment 1: The authors have addressed my questions and I would like to recommend publication in Nature Communications.

Response: We thank the reviewer for their thorough and timely evaluation of our manuscript and their recommendation to publish it in nature communication.